# CAPE: Generalized Convergence Prediction Across Architectures Without Full Training

**Alireza Pourali**  *alirezap@yorku.ca*
*Department of Electrical Engineering and Computer Science*
*York University*

**Arian Boukani**  *arbo@yorku.ca*
*Department of Electrical Engineering and Computer Science*
*York University*

**Hamzeh Khazaei**  *hkh@yorku.ca*
*Department of Electrical Engineering and Computer Science*
*York University*

**Reviewed on OpenReview:** *https://openreview.net/forum?id=wGngfOwBYn*

## Abstract

Training deep neural networks to convergence is expensive and time-consuming, especially when exploring new architectures or hardware configurations. Prior work has primarily estimated per-iteration or per-epoch cost under fixed training schedules, overlooking the critical challenge of predicting how long a model will take to converge. We present *CAPE* (Convergence-Aware Prediction Engine), a lightweight and probing-based framework that predicts the number of epochs required for convergence before any full training occurs. CAPE performs a brief probe at initialization using a small batch of data to extract analytical and dynamical features, including parameter count, dataset size, learning rate, batch size, gradient norm, Neural Tangent Kernel (NTK) trace, and initial loss. These features jointly characterize the model's optimization landscape and serve as input to a meta-model trained to forecast convergence horizons under a validation-based early-stopping criterion. CAPE achieves strong predictive correspondence to true convergence epochs, with a Pearson correlation of 0.89 across diverse architectures and datasets, demonstrating accurate and consistent convergence prediction across model families. By enabling zero-shot prediction of full-dataset convergence behaviour, CAPE provides a practical tool for rapid model selection, hyperparameter exploration, and resource-aware training planning.

## 1 Introduction

Deep neural networks (DNNs) require efficient training methods because their development depends on ample computational resources and extended training periods. The current techniques (Bergstra & Bengio, 2012; Li et al., 2018; Parker-Holder et al., 2020) require extensive experimental testing to find optimal hyperparameters and training schedules, which results in both time consuming delays and unnecessary resource consumption. The Neural Tangent Kernel (NTK) theory stands out as a significant contribution to understanding how deep neural networks converge in recent research (Arora et al., 2019; Jacot et al., 2018; Lee et al., 2019; Wang et al., 2022; Mu et al., 2020), which analyzes the training dynamics in high-dimensional settings. However, both theoretical and empirical works highlight critical limitations: NTK validity depends on tight rescaling conditions (Boix-Adserà & Littwin, 2023), extensions with regularization are still confined near initialization (Clerico & Guedj, 2024), and empirical scaling behaviours deviate significantly from NTK predictions (Vyas et al., 2023). These gaps limit applicability to real-world scenarios. Furthermore, studies on meta-learning (Ji et al., 2020; Ye et al., 2021; Chen et al., 2020; Guan et al., 2022; Harrison et al., 2022; Zhou

et al., 2019; Tack et al., 2022) aim to improve learning efficiency by leveraging knowledge from past tasks, but cannot often predict training time for held-out configurations (i.e., new model–dataset–hyperparameter tuples). Work on predicting computational cost (Justus et al., 2018; Pourali et al., 2025; Geoffrey et al., 2021) has shown promise, but often relies on linear models that fail to capture the inherent non-linearities of DNN training. To address these limitations, we introduce CAPE (Convergence-Aware Prediction Engine), a novel framework that combines analytical descriptors with probing-based feature extraction to accurately predict the number of epochs required for convergence under a validation-based early-stopping criterion.

Our proposed framework employs a lightweight, probing-based feature extraction method to characterize a DNN at initialization, eliminating the need for full training runs. This design draws inspiration from recent advances in linking initialization dynamics to optimization and generalization behavior (Wang et al., 2022; Ronen et al., 2019; Marion & Berthier, 2023). By extracting analytical and dynamical features such as the number of parameters, dataset size, learning rate, batch size, gradient norm, initial loss, and a proxy for the Neural Tangent Kernel (NTK) trace, CAPE captures both the structural complexity and early optimization smoothness of the model. These features form a meta-dataset that spans diverse neural architectures (MLPs, CNNs, RNNs, and Transformers) and datasets. The meta-dataset is then used to train a regression-based meta-model that learns the mapping between the extracted features and the number of epochs required for convergence, as determined by a validation-based early-stopping criterion. This meta-learning approach enables CAPE to generalize effectively to unseen architectures and datasets, providing accurate and efficient convergence predictions that surpass traditional analytical and linear regression baselines (Zancato et al., 2020; Kaplan et al., 2020; Hoffmann et al., 2022).

The key advantage of our approach lies in its ability to predict the number of epochs required for a model to reach convergence *before* initiating full training. This capability provides substantial benefits throughout the deep learning workflow. Researchers and practitioners can leverage these predictions to make informed choices regarding model selection, hyperparameter tuning, and the design of efficient training schedules. By accurately forecasting convergence behavior in advance, CAPE enables more effective allocation of computational resources and reduces the time spent on trial-and-error experimentation. The proposed framework overcomes the limitations of existing methods by combining probing-based feature extraction with the generalization capabilities of meta-learning, offering a practical and scalable solution for convergence prediction in deep neural networks. Ultimately, this work contributes to the broader goal of improving the efficiency and accessibility of deep learning research by allowing practitioners to focus on model design and evaluation rather than exhaustive training cycles.

To summarize, our contributions are the following:

(i) We introduce a probing-based convergence prediction framework that estimates the number of epochs required for deep neural networks (DNNs) to reach convergence, without executing full training runs.

(ii) We extract both structural and dynamical features, including parameter count, dataset size, batch size, learning rate, gradient norm, initial loss, and a proxy for the Neural Tangent Kernel (NTK) trace by probing the model at initialization using a small batch of data.

(iii) We construct a meta-dataset that spans diverse architectures (MLPs, CNNs, RNNs, Transformers) and datasets, capturing convergence behavior across a wide range of model configurations.

(iv) We develop a regression-based meta-model using a Random Forest ensemble, trained on the constructed meta-dataset to learn the complex mapping between initialization-time features and the number of epochs required for convergence under a validation-based early-stopping criterion. This meta-model generalizes effectively to held-out architectures and datasets not seen during meta-training.

(v) We demonstrate that our system enables early estimation of training cost, allowing researchers and practitioners to make informed decisions about model selection, resource budgeting, and training schedules, improving overall efficiency in deep learning workflows.

## 2 Related Works

### 2.1 Convergence Prediction in Deep Learning

Recent works on convergence analysis (Allen-Zhu et al., 2019; Zou et al., 2020; Ji et al., 2020; Gao et al., 2024; Zancato et al., 2020) in deep learning focus heavily on theoretical guarantees derived from over-parameterized networks and NTK-based formulations. These methods show that gradient descent can converge globally when networks are sufficiently wide and initialized under specific distributions, assuming either data separability or proximity to initialization. Beyond NTK, other lines of work establish convergence guarantees under weaker conditions such as gradient domination (Weissmann et al., 2025), logistic loss on two-layer networks (Gopalani et al., 2024), small initialization regimes (Kumar & Haupt, 2025), and optimizer-specific analyses of RMSProp/Adam (Zhang et al., 2025). For Transformers, convergence analysis has also considered implicit bias and self-attention dynamics (Vasudeva et al., 2024). Some models reformulate training dynamics as stochastic differential equations in function space, enabling closed form predictions of time-to-accuracy using the eigenvalues of the NTK (Zancato et al., 2020; Wan et al., 2021; Lee et al., 2019). However, these approaches are mostly confined to fine-tuning or pre-trained regimes, often requiring the network to remain in a small perturbation region around initialization (Allen-Zhu et al., 2019; Zancato et al., 2020). While such works provide valuable theoretical insights, their reliance on restrictive assumptions limits computational practicality and precludes advanced prediction of convergence before training begins, a gap our system directly addresses.

### 2.2 Learning Curve Extrapolation and Training Time Estimation

Learning curve extrapolation is widely used to estimate final performance from early training signals, especially within AutoML and neural architecture search workflows. Traditional approaches use Bayesian curve fitting, Gaussian processes, or ensemble-based estimators that require observing partial training trajectories (Domhan et al., 2015; Klein et al., 2017). More recent meta-learned models like LC-PFN (Adriaensen et al., 2023) reduce reliance on handcrafted priors and instead fit generalizable predictors over many tasks, enabling one-shot extrapolation of loss or accuracy. However, such methods still depend on partial curve data and are inapplicable when no training has occurred. Some frameworks like MOTE-NAS jointly predict resource use and model quality but focus more on cost or latency than convergence (Zhang et al., 2024b). Our work differs by offering zero-shot convergence prediction, estimating the number of epochs required for training to converge using initialization-only features, without relying on any portion of the learning curve (Zela et al., 2020; Bahri et al., 2024). Complementary studies have demonstrated that learning rate schedules, particularly cooldown phases, play a critical role in shaping convergence dynamics (Dremov et al., 2025), further motivating the need for predictors that generalize beyond handcrafted or schedule-dependent training trajectories.

### 2.3 Meta-Learning and Probing-Based Estimation

Meta-learning has proven effective for transferring training knowledge across tasks and model types, especially when applied to optimizer adaptation, initialization heuristics, and performance modeling (Ye et al., 2021; Zhang et al., 2024a; Guan et al., 2022). Convergence in distributed and federated settings has also been studied, where Local SGD exhibits accelerated rates for over-parameterized models (Qin et al., 2022), complementing meta-generalization perspectives. Some approaches use task-conditioned priors or learned regularizers that adapt to optimization landscapes dynamically, often leading to improved generalization across datasets and objectives (Tack et al., 2022; Jiang et al., 2021; Grant et al., 2018). Probing-based estimation complements this by extracting features like gradient norm, NTK trace proxies, and parameter counts at initialization to guide predictions about training dynamics (Zhu et al., 2022; Xia et al., 2020). These methods often avoid full training by leveraging small-batch statistics and meta-trained regressors, reducing computational overhead (Adriaensen et al., 2023; Wang & Ma, 2022). Despite their strengths, prior work typically focuses on few-shot learning or inner-loop optimization efficiency rather than directly predicting full-model convergence from scratch, which our work uniquely addresses.

## 3  Predicting Convergence Without Training

In this section, we formalize the task of predicting the convergence epochs of a deep neural network (DNN) without executing full training. Rather than relying on learning–curve extrapolation or empirical tuning, *CAPE* estimates the number of epochs required to reach a predefined convergence threshold, as determined by a validation-based early-stopping criterion. This estimation uses only analytical and dynamical features extracted at initialization, enabling zero-shot convergence prediction across architectures and datasets.

**Assumption 1 (Fixed and Reproducible Initialization).** *Model parameters $\theta \in \mathbb{R}^p$ are initialized with a deterministic scheme (e.g., Xavier or He) and a fixed random seed. All probing features are computed at this initialization point $\theta_0$; parameters remain unchanged during probing.*

Let $\mathcal{X}$ and $\mathcal{Y}$ denote the input and output spaces, respectively, and let the dataset be $\mathcal{D} = \{(x_i, y_i)\}_{i=1}^N \subset \mathcal{X} \times \mathcal{Y}$. We write the network as

$$f_\theta : \mathcal{X} \to \mathcal{Y}, \qquad \theta \in \mathbb{R}^p.$$

Given a task-appropriate loss $\ell : \mathcal{Y} \times \mathcal{Y} \to \mathbb{R}_{\geq 0}$, we define the empirical training and validation losses at epoch $t$ by

$$\mathcal{L}_{\text{train}}(t) \;=\; \frac{1}{|\mathcal{D}_{\text{train}}|} \sum_{(x,y) \in \mathcal{D}_{\text{train}}} \ell\big(f_{\theta^{(t)}}(x), y\big), \qquad \mathcal{L}_{\text{val}}(t) \;=\; \frac{1}{|\mathcal{D}_{\text{val}}|} \sum_{(x,y) \in \mathcal{D}_{\text{val}}} \ell\big(f_{\theta^{(t)}}(x), y\big),$$

where $\mathcal{D}_{\text{train}} \cup \mathcal{D}_{\text{val}} = \mathcal{D}$ is a fixed split and $\theta^{(t)}$ are the parameters after $t$ training epochs (used only to define the ground-truth label; CAPE does not execute these epochs at inference time).

**Definition (Convergence Epoch Count).** *The convergence epoch count $T_{\text{conv}}$ is defined as the smallest epoch at which the validation loss $\mathcal{L}_{\text{val}}(t)$ fails to improve by more than a tolerance $\delta$ for $p$ consecutive epochs:*

$$T_{\text{conv}} = \min \left\{ t \in \mathbb{N} \mid \mathcal{L}_{\text{val}}(t - p') - \mathcal{L}_{\text{val}}(t) < \delta, \; \forall p' \leq p \right\} \tag{1}$$

This criterion aligns with standard early-stopping practice in deep learning, replacing fixed loss ratios with a validation-based stopping rule.

**Assumption 2 (Expected Monotonic Decay).** *While stochastic optimizers such as SGD or Adam introduce noise in the loss trajectory, we assume that the* expected validation loss $\mathbb{E}[\mathcal{L}_{\text{val}}(t)]$ *decreases monotonically in expectation, ensuring that Eq. 1 yields a well-defined $T_{\text{conv}}$.*

To predict $T_{\text{conv}}$ without performing training, we extract a feature vector $\mathbf{z} \in \mathbb{R}^d$ from the model at initialization and from a small subset of the training data.

**Assumption 3 (Representative Probing Subset).** *A randomly sampled subset $\mathcal{D}_{probe} \subset \mathcal{D}$, with $|\mathcal{D}_{probe}| \ll N$, provides a sufficient approximation of the model's early-layer dynamics and curvature statistics, allowing the probing features $\mathbf{z}$ (defined in Section 4) to capture convergence-relevant behavior.*

**Probe-batch vs. full-dataset convergence.** All analytical and dynamical features (e.g., $\log \|\nabla \ell\|^2$, NTK trace, initial loss) are computed using the small subset $\mathcal{D}_{\text{probe}}$. However, the ground-truth convergence label $T_{\text{conv}}$ is measured on the full training dataset using the validation-based criterion in Eq. 1. This distinction enables *CAPE* to remain zero-shot at inference time while grounding its labels in realistic full-dataset behavior.

We then train a regression function $g : \mathbb{R}^d \to \mathbb{R}$ that maps the probing features to a predicted convergence epoch count $\hat{T}_{\text{conv}}$. A meta-dataset $\mathcal{M} = \{(\mathbf{z}^{(j)}, T_{\text{conv}}^{(j)})\}_{j=1}^M$ is constructed from diverse architectures, datasets, and training configurations. The meta-regressor is trained by minimizing the mean squared error in log space:

$$\min_{g \in \mathcal{G}} \frac{1}{M} \sum_{j=1}^M \left( \log g(\mathbf{z}^{(j)}) - \log T_{\text{conv}}^{(j)} \right)^2 \tag{2}$$

**Assumption 4 (Meta-Generalization).** *The meta-regressor $g$ trained on $\mathcal{M}$ generalizes to held-out architecture–dataset pairs drawn from the same underlying distribution.*

**Proposition 1 (Asymptotic Consistency).** *If the meta-dataset $\mathcal{M}$ is sufficiently diverse and the hypothesis class $\mathcal{G}$ is a rich function family (e.g., universal approximators), then as $M \to \infty$, the predictor $g$ converges in probability to the true mapping $\phi(f_{\theta_0}, \mathcal{D}_{probe}, \eta, B, N, a) \mapsto T_{\mathrm{conv}}$.*

This formulation enables convergence prediction in the early-stopping sense without performing full training, relying solely on computational and structural signals extracted at initialization.

## 4 Probing-Based Feature Extraction and Meta-Learning

CAPE's implementation comprises three stages: (1) designing initialization-time probing features that capture both structural and dynamical properties of a model; (2) constructing a diverse meta-dataset from multiple architectures and datasets; and (3) training a meta-regressor to predict convergence epochs from these features. This section details each stage of the system.

### 4.1 Probing Feature Design

We extract a compact set of features from a randomly initialized model using a small probe subset of the training data. As stated in Assumption 1, initialization is fixed and reproducible, and no parameter updates are performed, making the procedure efficient and architecture-agnostic.

Let $f_{\theta_0}$ denote the neural network at initialization with parameters $\theta_0 \in \mathbb{R}^p$, and let $\mathcal{D}_{\mathrm{probe}} \subset \mathcal{D}$ be a randomly sampled subset with size $B \ll |\mathcal{D}|$, as described in Assumption 3. The probing function is defined as

$$\mathbf{z} = \phi(f_{\theta_0}, \mathcal{D}_{\mathrm{probe}}, \eta, B, N, a), \tag{3}$$

where $\eta$ is the learning rate, $B$ is the batch size, $N$ is the total dataset size, and $a$ is an architecture identifier (e.g., MLP, CNN, RNN, Transformer). Each element influences the expected optimization dynamics. The following initialization-time features are computed:

**Parameter Count $P$.** Measures the representational complexity of the model, reflecting its total number of learnable parameters:

$$P = |\theta_0|. \tag{4}$$

**Initial Loss $\mathcal{L}_0$.** Quantifies the model's empirical loss at random initialization, reflecting how well untrained features align with the target distribution:

$$\mathcal{L}_0 = \frac{1}{B} \sum_{(x,y) \in \mathcal{D}_{\mathrm{probe}}} \ell(f_{\theta_0}(x), y). \tag{5}$$

It reflects the alignment between random features and target distributions.

**Average Gradient Norm $G^2$.** Sensitivity of the loss to parameter updates at initialization:

$$G^2 = \frac{1}{B} \sum_{(x,y) \in \mathcal{D}_{\mathrm{probe}}} \left\| \nabla_{\theta_0} \ell(f_{\theta_0}(x), y) \right\|^2. \tag{6}$$

Larger values typically indicate a steeper loss surface and faster initial descent, whereas smaller values suggest flat or ill-conditioned regions.

**NTK Trace Proxy $\tau$.** A curvature-sensitive quantity capturing how model outputs vary with parameter perturbations:

$$\tau = \frac{1}{B} \sum_{x \in \mathcal{D}_{\mathrm{probe}}} \left\| \nabla_{\theta_0} f_{\theta_0}(x) \right\|^2. \tag{7}$$

**Log-Transformed Feature Vector.** To reduce scale variance and improve numerical stability, all scalar quantities are log-transformed before regression:

$$\mathbf{z} = \big[\log(P), \log(\mathcal{L}_0), \log(G^2), \log(\tau), \log(N), \log(\eta), \log(B), a\big]. \tag{8}$$

All features can be computed within a second on modern hardware, allowing large-scale or real-time evaluation.

## 4.2 Meta-Dataset Construction

For each model–dataset configuration, we pair the extracted features $\mathbf{z}^{(j)}$ with the ground-truth convergence epoch count $T_{\text{conv}}^{(j)}$ defined by Eq. 1. The resulting meta-dataset is

$$\mathcal{M} = \big\{(\mathbf{z}^{(j)}, T_{\text{conv}}^{(j)})\big\}_{j=1}^{M}, \tag{9}$$

spanning diverse architectures, datasets, and training conditions to promote generalization.

## 4.3 Meta-Regressor Training

Given $\mathcal{M}$, we train a regression model $g : \mathbb{R}^d \to \mathbb{R}$ that maps probing features to predicted convergence epochs. We employ a Random Forest regressor (Breiman, 2001) due to its robustness to feature scale, interpretability, and strong empirical performance on heterogeneous feature sets. The model is trained on log-transformed inputs and targets to reduce scale sensitivity and stabilize variance across configurations, minimizing the objective in Eq. 2. Regularization is implicitly achieved through bootstrap aggregation and random feature sampling, while model selection and hyperparameter tuning are performed via $k$-fold cross-validation on the training meta-set.

# 5 Experiments

We evaluate *CAPE* across four architecture families such as MLPs, CNNs, RNNs, and Transformers, each instantiated using compact yet representative models to capture diverse convergence behaviors. All experiments follow a unified probing and validation-based convergence framework, ensuring comparability across architectures.

**Model families and training protocol.** Each model family is represented by widely used architectures scaled for efficient convergence measurement. MLP-based models include *MLP-Mixer* (Tolstikhin et al., 2021), *ResMLP* (Touvron et al., 2022), and *AS-MLP* (Lian et al., 2021). CNNs are represented by *ResNet-50* (He et al., 2016), *DenseNet-121* (Huang et al., 2017), and *MobileNetV2* (Sandler et al., 2018). RNNs include *LSTM* (Hochreiter & Schmidhuber, 1997), *GRU* (Cho et al., 2014), and *BiLSTM* (Huang et al., 2015), while Transformers cover *DeiT-Tiny* (Touvron et al., 2021) and *DistilBERT* (Sanh et al., 2019). All models are trained under a consistent hyperparameter grid with learning rates $LR \in \{5 \times 10^{-4}, 10^{-3}, 2 \times 10^{-3}\}$, batch sizes $B \in \{8, 16, 32, 64, 128, 256\}$, and optimizers *Adam*, *Adafactor*, *AdamW* and *SGD*.

**Datasets.** We evaluate each model on the datasets that align with its standard benchmark usage to ensure architectural relevance and consistent convergence characteristics:

- **MLPs:** AS-MLP and MLP-Mixer are evaluated using TinyImageNet (Deng et al., 2009) and STL10 (Coates et al., 2011); ResMLP is evaluated using CIFAR-100 (Wei et al., 2021) and TinyImageNet.

- **CNNs:** ResNet-50 is evaluated using TinyImageNet and CIFAR-10 (Wei et al., 2021); DenseNet-121 on CIFAR-100 and TinyImageNet; and MobileNetV2 on CIFAR-10 and STL10.

- **RNNs:** GRU and LSTM are evaluated using IMDB (Maas et al., 2011) and AG NEWS (Gulli, 2005); BiLSTM on SST2 (Socher et al., 2013) and IMDB.

- **Transformers:** DistilBERT is evaluated using SST2 and IMDB; DeiT-Tiny on CIFAR-100 and TinyImageNet.

All datasets are preprocessed following standard normalization and tokenization protocols. Image datasets are resized to their canonical benchmark resolutions consistent with prior work (e.g., CIFAR and Tiny-ImageNet), while text datasets use subword or whitespace tokenization with vocabularies capped at 30k tokens.

**Meta-feature extraction.** To estimate the convergence epoch $T_{\text{conv}}$ from initialization, we extract six probing features from a single training batch per model–dataset pair: parameter count ($\log P$), learning rate ($\log \text{LR}$), batch size ($\log B$), squared gradient norm ($\log G^2$), NTK trace proxy ($\log \tau$), and initial loss ($\log L_0$). For each configuration, the ground-truth label $T_{\text{conv}}$ is defined as the first epoch at which the validation loss fails to improve by at least the chosen tolerance threshold ($5 \times 10^{-4}$) for five consecutive epochs. This tolerance level aligns with widely used early-stopping practices in modern deep learning, where improvements below the order of $10^{-3}$ to $10^{-4}$ are typically treated as numerically insignificant and attributed to routine stochastic variation. All probing features were computed from a single mini-batch corresponding to the smallest batch size defined for each model–dataset configuration. Although this represents a minimal subset of the training data, CAPE consistently achieved accurate convergence predictions across architectures and datasets, underscoring the robustness of its initialization-time feature extraction.

We train a Random Forest regressor to predict $\log T_{\text{conv}}$ from these features, using 200 estimators, maximum depth of 8, bootstrap sampling, and mean squared error as the split criterion. Predictions are exponentiated to recover $T_{\text{conv}}$ in epoch units. We choose Random Forest for its robustness to noisy features, low sensitivity to scaling, and strong performance on heterogeneous tabular datasets.

**Hardware and runtime.** All experiments were conducted on a workstation equipped with an NVIDIA RTX 4000 Ada GPU, an Intel Core i9-14900K CPU, and 64 GB of RAM. The full meta-dataset generation, covering all model families (MLPs, CNNs, RNNs, and Transformers), multiple datasets, and exhaustive sweeps over learning rates, batch sizes, optimizers, required over 400 GPU hours of compute time.

**Evaluation protocol.** We evaluate CAPE under three out-of-sample regimes that mirror realistic deployment: (i) *5-fold cross validation (CV)* with shuffled folds; (ii) *Leave-One-Dataset-Out (LODO)*, where each dataset is held out in turn; and (iii) *Leave-One-Model-Out (LOMO)*, where each model family is held out in turn. For every regime, we generate strictly out-of-fold predictions: in CV, a standard 5-fold cross-validation procedure is applied, whereas in LODO and LOMO, a Leave-One-Group-Out splitter grouped by dataset or model is used. To ensure reproducibility, the code and datasets used in our experimental evaluation are publicly available in our GitHub repository[1].

**Baselines.** To the best of our knowledge, no prior work has demonstrated *zero-shot* prediction of the number of training epochs $T_{\text{conv}}$ required for convergence under a validation-based early-stopping criterion, for a novel combination of architecture, dataset, and hyperparameters, using only single batch probing features and a learned regressor. An ablation of alternative regressors for CAPE and baseline models is reported in Appendix A.3. We compare against three representative baselines, both parametric and non-parametric, each evaluated under the same out-of-sample protocol:

(1) *CAPE (probe-only)* trains an identical Random Forest but *only* on the two probe features $\log G^2$ and $\log \tau$.

(2) Learning-Curve Extrapolation *(LCE)* (Domhan et al., 2015) predicts $T_{\text{conv}}$ from an exponential fit to the validation loss prefix; the decay/offset hyperparameters are learned *within each training fold* and the amplitude is calibrated on the test prefix before applying the early-stopping rule.

(3) *Scaling-Law* uses a ridge regressor on $\{\log P, \log N\}$ only.

---

[1] https://github.com/pacslab/CAPE

Table 1: CAPE vs. baselines across evaluation protocols. Lower MAE/RMSE and higher PearsonR indicate better performance.

| Method | Evaluation protocol | MAE | RMSE | PearsonR |
|---|---|---|---|---|
| **CAPE (full feature set)** | | | | |
| CAPE | Cross-fold (5-fold) | **4.63** | **8.10** | **0.89** |
| CAPE | Leave-one-dataset-out | **6.85** | **10.57** | **0.81** |
| CAPE | Leave-one-model-out | **7.27** | **11.04** | **0.79** |
| **Baselines** | | | | |
| Probe-only $(\log G^2, \log \tau)$ | Cross-fold (5-fold) | 13.10 | 21.33 | 0.05 |
| Probe-only $(\log G^2, \log \tau)$ | Leave-one-dataset-out | 13.90 | 21.69 | $-0.09$ |
| Probe-only $(\log G^2, \log \tau)$ | Leave-one-model-out | 15.12 | 23.12 | $-0.16$ |
| Learning-curve extrapolation | Cross-fold (5-fold) | 13.53 | 29.22 | 0.64 |
| Learning-curve extrapolation | Leave-one-dataset-out | 16.09 | 33.68 | 0.65 |
| Learning-curve extrapolation | Leave-one-model-out | 14.43 | 31.08 | 0.66 |
| Scaling-law $(\log P, \log N)$ | Cross-fold (5-fold) | 11.87 | 17.23 | 0.28 |
| Scaling-law $(\log P, \log N)$ | Leave-one-dataset-out | 11.81 | 17.29 | 0.27 |
| Scaling-law $(\log P, \log N)$ | Leave-one-model-out | 12.34 | 17.82 | 0.18 |

**Metrics.** We report three complementary metrics for CAPE: Mean Absolute Error (MAE), Root Mean Squared Error (RMSE) and Pearson correlation coefficient ($r$). For the baselines we report MAE, RMSE, and $r$. All metrics are computed on the out-of-fold predictions produced by the corresponding regime.

- **Mean Absolute Error (MAE).** Measures the average absolute difference between the predicted and actual convergence epochs:

$$\text{MAE} = \frac{1}{n} \sum_{i=1}^{n} |T_{\text{pred},i} - T_{\text{act},i}|.$$

- **Root Mean Squared Error (RMSE).** Emphasizes larger prediction deviations by squaring the residuals before averaging:

$$\text{RMSE} = \sqrt{\frac{1}{n} \sum_{i=1}^{n} (T_{\text{pred},i} - T_{\text{act},i})^2}.$$

- **Pearson Correlation Coefficient ($r$).** Quantifies the linear correlation between predicted and actual convergence epochs:

$$r = \frac{\sum_{i=1}^{n} (T_{\text{pred},i} - \overline{T}_{\text{pred}})(T_{\text{act},i} - \overline{T}_{\text{act}})}{\sqrt{\sum_{i=1}^{n} (T_{\text{pred},i} - \overline{T}_{\text{pred}})^2 \sum_{i=1}^{n} (T_{\text{act},i} - \overline{T}_{\text{act}})^2}}.$$

## 5.1 Comparison with Baselines

**Evaluation and comparison.** Across all protocols, CAPE attains substantially lower absolute errors and markedly higher correlation than the baselines (Table 1). In cross-fold evaluation, CAPE achieves MAE 4.63 and RMSE 8.10 with Pearson correlation 0.89. When holding out entire datasets or models, CAPE maintains strong accuracy (MAE 6.85/7.27; RMSE 10.57/11.04) and robust rank correlation (PearsonR 0.81/0.79). The probe-only variant, which removes all static/context features, shows large errors and near-zero (or negative) correlations, indicating that probe signals alone are insufficient. Learning-curve extrapolation, despite consuming early validation prefixes, underperforms CAPE on both error and correlation, highlighting the advantage of CAPE's initialization-time features coupled with meta-regression. The simple scaling-law baseline captures coarse trends from $\{\log P, \log N\}$ but lacks the fidelity to match CAPE's accuracy or

correlation. Collectively, these results demonstrate that CAPE's integrated feature design and training protocol provide reliable, generalizable convergence-epoch predictions under diverse evaluation regimes.

**Relation to neural scaling laws.** Neural scaling laws posit approximate power-law relationships between model/dataset scale and performance or compute (Kaplan et al., 2020; Hoffmann et al., 2022). Our scaling-law baseline follows this spirit by regressing convergence epochs on $\{\log P, \log N\}$. While such macroscopic laws capture broad monotone trends, they deliberately abstract away optimization specifics (e.g., learning rate, batch size, curvature/gradient geometry). By contrast, CAPE augments static scale with initialization-time probes $(\log G^2, \log \tau)$ and training-context features, enabling it to account for short-horizon dynamics that materially affect when early stopping is triggered. Empirically, this yields substantially tighter fit (lower MAE/RMSE) and stronger correspondence with ground truth than scale-only predictors, indicating that convergence timing is not solely a function of $(P, N)$ but also of local geometry and optimizer scale.

**Scaling-law versus Probe-based predictors.** Scaling-law estimators are most effective in the *pre-training design phase*, where only coarse descriptors, such as model parameter count and dataset size are known. They provide inexpensive and interpretable approximations of asymptotic behavior, allowing practitioners to forecast compute requirements or performance trends under fixed training protocols. However, because these estimators assume a stationary training regime and abstract away optimization dynamics, they often fail to capture non-scaling effects introduced by changes in hyperparameters, architectural inductive biases, or data distributions. In contrast, CAPE operates at the *initialization and probing level*, integrating both structural and early-dynamic signals (e.g., gradient norm, NTK trace, optimizer configuration) to model short-horizon convergence behavior. This makes CAPE particularly suitable when the training setup deviates from canonical scaling assumptions, for instance, when adjusting batch size or learning rate, employing adaptive optimizers, or applying early stopping criteria. In such regimes, CAPE consistently achieves lower prediction error and higher agreement with realized convergence compared to scale-only baselines, offering a more robust and context-aware characterization of training dynamics while maintaining lightweight computational overhead.

**Comparison with learning-curve extrapolation.** Our LCE baseline fits an exponential model $L(t) = ae^{-bt} + c$ to the initial portion of each run's validation-loss trajectory. The decay and asymptote parameters $(b, c)$ are estimated from the training data and applied to held-out runs, after which the fitted curve is extrapolated to the full training horizon and an early-stopping rule is simulated using each run's patience and tolerance parameters. Both CAPE and LCE predict the same convergence target and are evaluated under identical experimental protocols with shared metrics. Across all settings, CAPE achieves substantially lower MAE and RMSE and higher correlation than LCE (Table 1). This result suggests that early validation trajectories, though informative in principle, are often noisy and sensitive to optimization hyperparameters, whereas CAPE's initialization-time geometric probes and contextual features yield more stable and generalizable convergence predictions.

## 5.2 Cross-Architecture and Cross-Dataset Convergence Prediction

**Evaluation Protocol.** As shown in Table 2, we report per-model performance of CAPE under three evaluation regimes: 5-fold cross-validation (CV), Leave-One-Dataset-Out (LODO), and Leave-One-Model-Out (LOMO). Each entry summarizes mean absolute error (MAE) and root mean squared error (RMSE) in epochs.

**Observations.** CAPE maintains low prediction error (typically 2–8 epochs) across most architectures and evaluation regimes. Recurrent families (BiLSTM, GRU, LSTM) and several CNN/MLP variants (DenseNet121, ResMLP, Mixer) show stable performance, indicating that the probe-based features generalize well beyond the specific training domains. Interestingly, CAPE occasionally performs better under cross-domain regimes than in standard cross-validation. For example, DeiT-Tiny and DistilBERT achieve lower MAE and RMSE under LODO, indicating that dataset-level generalization can benefit Transformers whose convergence dynamics are relatively stable across datasets with similar optimization profiles. Likewise, the LSTM model attains slightly lower error under LOMO, suggesting that recurrent architectures

Table 2: Per-model convergence prediction for CAPE across evaluation regimes. Lower is better for MAE/RMSE (epochs).

| Model | CV (5-fold) | | LODO | | LOMO | |
|---|---|---|---|---|---|---|
| | MAE | RMSE | MAE | RMSE | MAE | RMSE |
| DeiT-Tiny | 4.42 | 5.72 | 3.84 | 4.99 | 4.90 | 6.28 |
| DistilBERT | 2.34 | 3.37 | 1.79 | 2.55 | 2.95 | 3.84 |
| AS-MLP | 12.63 | 16.23 | 14.46 | 18.64 | 19.30 | 23.05 |
| BiLSTM | 5.21 | 9.59 | 6.84 | 12.98 | 5.74 | 10.65 |
| DenseNet121 | 2.20 | 3.50 | 4.25 | 4.94 | 3.58 | 4.31 |
| GRU | 2.67 | 4.40 | 7.22 | 10.15 | 5.28 | 7.45 |
| LSTM | 5.19 | 9.20 | 8.89 | 13.13 | 4.86 | 8.63 |
| MLP-Mixer | 5.25 | 11.18 | 5.99 | 11.37 | 6.51 | 10.07 |
| MobileNetV2 | 4.94 | 6.14 | 11.23 | 12.01 | 9.81 | 12.11 |
| ResMLP | 3.56 | 5.67 | 4.49 | 7.12 | 4.99 | 8.29 |
| ResNet-50 | 2.53 | 3.28 | 6.31 | 7.56 | 12.07 | 13.29 |

share transferable temporal-gradient statistics that help CAPE interpolate across unseen sequence models. In contrast, convolutional and MLP-based families (DenseNet121, ResNet-50, ResMLP, Mixer) depend more strongly on dataset-specific features such as input resolution, normalization depth, and augmentation scheme, causing CV to remain their most favorable regime. Finally, AS-MLP exhibits the largest MAE/RMSE due to its broad convergence range (26–67 epochs), where proportionally similar relative errors translate into large absolute deviations in epoch counts. Moreover, LOMO remains the most challenging regime overall, as it requires extrapolation to unseen architectures; nevertheless, CAPE maintains competitive MAE (often <10 epochs) across most model families, demonstrating its ability to capture transferable convergence signatures across both datasets and architectures.

## 5.3 Cross-Dataset Behavior under Batch-Size Variation

We evaluate CAPE under the Leave-One-Dataset-Out (LODO) protocol while systematically sweeping key hyperparameters, batch size, learning rate, and optimizer type, to examine its robustness to training configuration shifts. For each model, training is performed on all available datasets except one, which is reserved for evaluation. Specifically, DeiT-Tiny, ResNet-50, DenseNet-121, MLP-Mixer, and ResMLP are tested on TinyImageNet, DistilBERT and GRU on IMDB, and BiLSTM on SST-2 as the unseen datasets. Figure 1 summarizes CAPE's Leave-One-Dataset-Out performance across varying batch sizes for all model families. Overall, predicted convergence epochs remain closely aligned with the actual values, indicating that the meta-regressor generalizes well across unseen datasets. For recurrent models (BiLSTM, GRU), CAPE slightly overestimates convergence at smaller training batch sizes but achieves stable agreement as batch size increases, reflecting that probe-derived features computed using the smallest batch size per model–dataset configuration, captures optimization dynamics robustly despite changes in batch-dependent gradient noise during training. Transformer-based architectures (DeiT-Tiny, DistilBERT) show consistently accurate predictions with only mild variability as batch size grows, suggesting that probe statistics such as gradient norm and NTK trace effectively encode convergence behavior largely independent of mini-batch scale. Among convolutional and MLP-based architectures (DenseNet-121, MLP-Mixer, ResMLP, ResNet-50), predicted values closely track actual convergence trends, with minor underestimation observed for ResMLP and ResNet-50 at intermediate batch sizes. These deviations likely arise from dataset-specific optimization effects on Tiny-ImageNet, where higher input variability and feature diversity slightly delay convergence relative to the probe-based expectation.

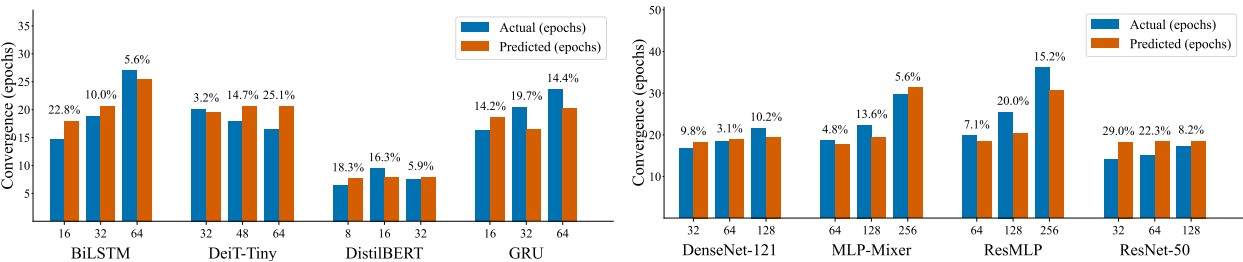

Figure 1: LODO evaluation across batch-size variations. Predicted versus actual convergence epochs are shown for each model, with relative errors (%) indicated above each configuration.

## 5.4 Cross-Dataset Behavior under Learning-Rate Variation

Figure 2 presents CAPE's Leave-One-Dataset-Out evaluation across different learning rates, illustrating the model's robustness to optimization-scale variation. Overall, the predicted and actual convergence epochs remain closely aligned, with most relative errors below 15%, confirming that CAPE effectively models convergence dynamics across a broad range of learning rates. For recurrent architectures (BiLSTM, GRU), prediction accuracy slightly degrades at intermediate learning rates, likely due to increased gradient oscillation that perturbs early probe statistics. Transformer models (DeiT-Tiny, DistilBERT) exhibit highly stable predictions at lower and moderate rates, with minor overestimation appearing only at the highest setting, indicating that probe-derived features generalize well across smooth optimization regimes. Convolutional and MLP-based models (DenseNet-121, MLP-Mixer, ResMLP, ResNet-50) maintain strong agreement overall, with mild over-prediction observed at elevated learning rates, particularly for ResMLP and ResNet-50, where faster initial loss decay leads to slightly premature predicted convergence. These results demonstrate that CAPE generalizes reliably to unseen datasets under varying learning-rate scales, preserving accurate convergence estimation across diverse model families.

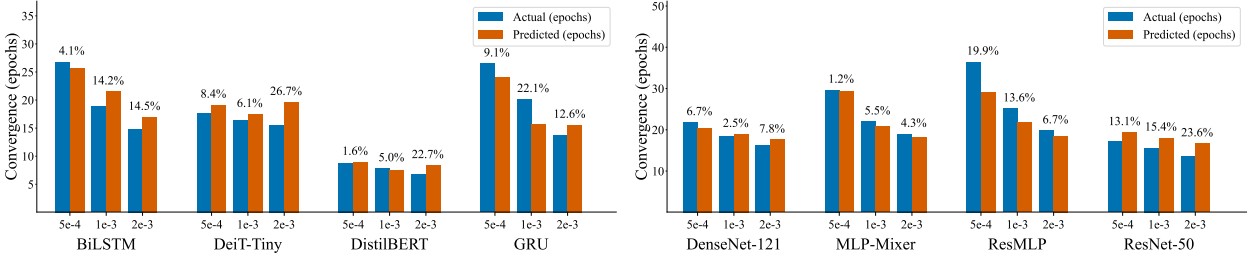

Figure 2: LODO evaluation across learning-rate variations. Configurations correspond to typical values such as $5 \times 10^{-4}$, $10^{-3}$, and $2 \times 10^{-3}$, with predicted versus actual convergence epochs compared for each model.

## 5.5 Cross-Dataset Behavior under Optimizer Variation

Figure 3 illustrates CAPE's Leave-One-Dataset-Out results across optimizer variations, evaluating the system's robustness to optimization dynamics. Across all architectures, predicted convergence epochs remain closely aligned with actual values, confirming CAPE's ability to generalize across distinct update rules. Recurrent models (BiLSTM, GRU) show stable predictions with deviations typically under 10% when switching between AdamW and SGD, while Transformer architectures (DeiT-Tiny, DistilBERT) maintain low error across Adam, AdamW, and Adafactor, indicating that probe features effectively encode optimizer-invariant characteristics of early training dynamics. Across all models, CAPE accurately reflects the longer convergence trajectories characteristic of SGD compared to adaptive optimizers such as AdamW. This behavior demonstrates that CAPE not only generalizes across optimizers but also faithfully captures optimizer-dependent convergence patterns across diverse architectures.

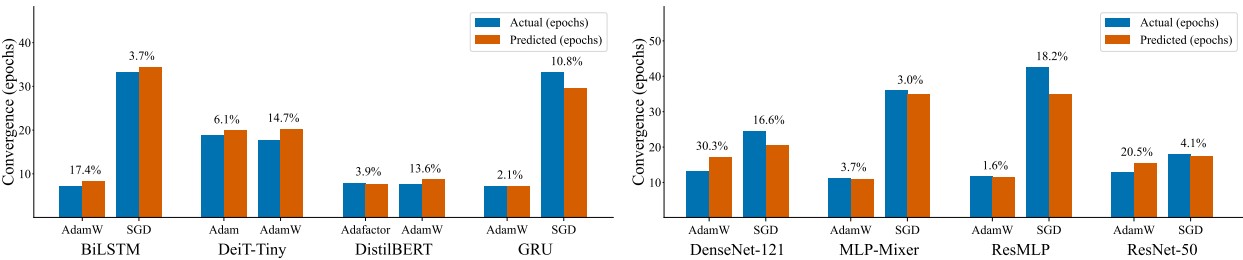

Figure 3: LODO evaluation across optimizer variations. Each model family is tested with two optimizers (e.g., AdamW and SGD for CNNs/MLPs, Adam and AdamW for Transformers, Adafactor and AdamW for sequence models), comparing predicted and actual convergence epochs.

# 6 Conclusion and Limitation

We introduced CAPE, a lightweight, probing-based framework that predicts the *epochs to convergence* without full training. By combining structural descriptors (e.g., parameter and dataset scale, batch size, learning rate) with initialization-time probes (gradient-norm statistics, NTK-trace proxy, initial loss), CAPE learns a meta-regressor that generalizes across architectures and datasets under a validation-based early-stopping rule. Empirically, CAPE achieves low absolute error and strong linear correspondence to ground truth, with Pearson correlation of 0.89 in cross-fold evaluation and remaining robust under LODO and LOMO protocols. Across MLPs, CNNs, RNNs, and Transformers, CAPE consistently outperforms static scale-only, prefix-based (LCE), and probe-only variants, enabling practical *ex-ante* convergence estimation for model selection, hyperparameter exploration, and compute planning.

**Hardware and runtime mapping.** While CAPE is hardware-agnostic by predicting epochs rather than wall-clock time, deployment often requires translating epochs into runtime or cost. Augmenting the feature set with hardware factors (e.g., GPU architecture/memory bandwidth, vCPU count, RAM, storage) would enable end-to-end time and cost prediction.

**Meta-dataset coverage.** Accuracy depends on the diversity of meta-training data. Broader coverage—additional modalities, longer sequences, larger image resolutions, and stronger augmentation regimes—should further improve robustness under distribution shift.

**Scope of probes and criteria.** Probes are computed at initialization and labels follow a specific early-stopping criterion. Extending to schedule-aware settings (e.g., cosine/step decay, cooldown phases) and alternative stopping rules could increase fidelity in scenarios with non-monotone trajectories.

**Extrapolation limits.** Although CAPE generalizes well to unseen datasets and model families, extreme regimes (very large/small learning rates or batch sizes, unusual optimizers) may require targeted meta-data or mild task-specific calibration.

**Modality-dependent convergence behavior.** The current early-stopping rule applies a patience of five consecutive epochs, which aligns well with the training dynamics of the vision datasets included in our meta-dataset. However, the notion of an "epoch" varies across modalities: text datasets often contain fewer samples, exhibit tokenization and sequence length dependent variability, and may complete an epoch with substantially fewer effective parameter update cycles than vision tasks. Developing a modality aware convergence definition, potentially adjusting patience thresholds or stabilization criteria to reflect dataset granularity would require redefining the convergence unit, regenerating the meta-dataset, and recalibrating all evaluation metrics to preserve architectural consistency. We leave this to future work, as exploring modality specific convergence signals, particularly for NLP workloads, represents a promising direction for expanding the generality of CAPE.

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

# A    Appendix

## A.1    Proof of Proposition 1: Asymptotic Consistency

**Proposition 1 (Asymptotic Consistency).** Assume that:

- The probing feature extractor $\varphi(f_{\theta_0}, \mathcal{D}_{\text{probe}})$ produces bounded feature vectors $\mathbf{z} \in \mathbb{R}^d$,

- The ground-truth mapping $h^*(\mathbf{z}) := T^*_{\text{conv}}$ is measurable,

- The hypothesis class $\mathcal{G}$ contains a sequence of functions that can approximate $h^*$ arbitrarily well (i.e., $\mathcal{G}$ is dense in $L^2(P_Z)$),

- The training examples $(\mathbf{z}^{(j)}, T^{*(j)}_{\text{conv}})$ are i.i.d. samples from a fixed distribution $\mathcal{D}_{\mathcal{M}}$.

Then, the empirical risk minimizer

$$g_M \in \arg\min_{g \in \mathcal{G}} \frac{1}{M} \sum_{j=1}^{M} \left( \log g(\mathbf{z}^{(j)}) - \log T^{*(j)}_{\text{conv}} \right)^2$$

satisfies

$$\lim_{M \to \infty} \mathbb{E}_{(\mathbf{z}, T^*_{\text{conv}})} \left[ (\log g_M(\mathbf{z}) - \log T^*_{\text{conv}})^2 \right] = \inf_{g \in \mathcal{G}} \mathbb{E}_{(\mathbf{z}, T^*_{\text{conv}})} \left[ (\log g(\mathbf{z}) - \log T^*_{\text{conv}})^2 \right].$$

That is, $g_M$ is a consistent estimator of the log-space convergence predictor.

**Proof.** Let $\mathcal{Z} = \mathbb{R}^d \times \mathbb{R}_{>0}$, where each sample $(\mathbf{z}, T^*_{\text{conv}}) \sim \mathcal{D}_{\mathcal{M}}$. Define the per-example loss function as

$$L(g; \mathbf{z}, T^*_{\text{conv}}) = (\log g(\mathbf{z}) - \log T^*_{\text{conv}})^2,$$

assuming $g(\mathbf{z}) > 0$ to ensure the logarithm is well-defined.

Define the population risk:

$$\mathcal{R}(g) := \mathbb{E}_{(\mathbf{z}, T^*_{\text{conv}}) \sim \mathcal{D}_{\mathcal{M}}} [L(g; \mathbf{z}, T^*_{\text{conv}})],$$

and the empirical risk over $M$ samples:

$$\hat{\mathcal{R}}_M(g) := \frac{1}{M} \sum_{j=1}^{M} L(g; \mathbf{z}^{(j)}, T^{*(j)}_{\text{conv}}).$$

Let $g_M \in \arg\min_{g \in \mathcal{G}} \hat{\mathcal{R}}_M(g)$. We aim to show that

$$\mathcal{R}(g_M) \to \inf_{g \in \mathcal{G}} \mathcal{R}(g) \quad \text{as } M \to \infty.$$

**Step 1: Uniform Convergence.** If the loss class $\mathcal{L}_{\mathcal{G}} := \{(\mathbf{z}, T^*_{\text{conv}}) \mapsto L(g; \mathbf{z}, T^*_{\text{conv}}) \mid g \in \mathcal{G}\}$ has finite pseudo-dimension or bounded covering number, then by the uniform law of large numbers:

$$\sup_{g \in \mathcal{G}} \left| \hat{\mathcal{R}}_M(g) - \mathcal{R}(g) \right| \xrightarrow{P} 0 \quad \text{as } M \to \infty.$$

This holds if $g$ is Lipschitz (e.g., tree or neural regressors with bounded weights), and both $\log T^*_{\text{conv}}$ and $\log g(\mathbf{z})$ are bounded, e.g., via clipping or regularization.

**Step 2: Approximation Error.** By assumption, there exists $g^* \in \mathcal{G}$ such that $\mathcal{R}(g^*) = \inf_{g \in \mathcal{G}} \mathcal{R}(g)$. Therefore, $\mathcal{G}$ can approximate the Bayes-optimal predictor arbitrarily closely.

**Step 3: Conclude Consistency.** From uniform convergence and richness of $\mathcal{G}$, we have:

$$\lim_{M \to \infty} \mathcal{R}(g_M) = \inf_{g \in \mathcal{G}} \mathcal{R}(g),$$

which implies the consistency of the meta-regressor:

$$\lim_{M \to \infty} \mathbb{E}_{(\mathbf{z}, T^*_{\text{conv}})} \left[ (\log g_M(\mathbf{z}) - \log T^*_{\text{conv}})^2 \right] = \inf_{g \in \mathcal{G}} \mathcal{R}(g).$$

## A.2 Supporting Lemma: Generalization of Empirical Risk Minimization

**Lemma 1 (ERM Convergence).** Let $\mathcal{G}$ be a hypothesis class of real-valued functions mapping from $\mathbb{R}^d$ to $\mathbb{R}_{>0}$, and assume that:

- The per-sample loss is $L(g; \mathbf{z}, T^*_{\text{conv}}) = (\log g(\mathbf{z}) - \log T^*_{\text{conv}})^2$,

- The examples $(\mathbf{z}, T^*_{\text{conv}})$ are i.i.d. from a fixed distribution $\mathcal{D}_{\mathcal{M}}$,

- The function class $\mathcal{G}$ has finite VC-dimension or bounded Rademacher complexity,

- $\log g(\mathbf{z})$ and $\log T^*_{\text{conv}}$ are uniformly bounded almost surely.

Then, the empirical risk minimizer

$$g_M = \arg\min_{g \in \mathcal{G}} \frac{1}{M} \sum_{j=1}^{M} \left( \log g(\mathbf{z}^{(j)}) - \log T^{*(j)}_{\text{conv}} \right)^2$$

satisfies, with high probability,

$$\left| \mathbb{E}[L(g_M; \mathbf{z}, T^*_{\text{conv}})] - \inf_{g \in \mathcal{G}} \mathbb{E}[L(g; \mathbf{z}, T^*_{\text{conv}})] \right| \le \epsilon(M),$$

where $\epsilon(M) \to 0$ as $M \to \infty$.

**Proof.** This follows from standard learning theory results. Since the loss is bounded and Lipschitz in $\log g(\mathbf{z})$, and $\mathcal{G}$ has finite capacity, uniform convergence holds over $\mathcal{L}_{\mathcal{G}} = \{L(g; \cdot, \cdot) : g \in \mathcal{G}\}$:

$$\sup_{g \in \mathcal{G}} \left| \hat{\mathcal{R}}_M(g) - \mathcal{R}(g) \right| \to 0 \quad \text{as } M \to \infty.$$

Then, by consistency of ERM under uniform convergence, the excess risk of $g_M$ over the best-in-class predictor vanishes:

$$\mathcal{R}(g_M) - \inf_{g \in \mathcal{G}} \mathcal{R}(g) \to 0.$$

## A.3 Ablation Study on Regressor Choices

To justify the regression models used for CAPE and the baselines, we present a comprehensive ablation over three regression families of *Ridge*, *Random Forest*, and *Gradient Boosting*. Table 3 summarizes performance across 5-fold CV, LODO, and LOMO regimes using MAE, RMSE, and Pearson correlation.

**Regressor choice for CAPE.** For CAPE, which incorporates both structural descriptors and probing-based features, non-linear tree-based regressors (Random Forest and Gradient Boosting) consistently out-perform Ridge across all regimes: they reduce MAE by 35–45% in cross-fold CV and by 25–40% in LODO, while also yielding substantially higher Pearson correlations. A similar pattern holds for the CAPE (probe-only) variant: although it operates solely on $\log G^2$ and $\log \tau$, tree-based models still achieve lower prediction error than Ridge, indicating that even the probe features exhibit non-linear relationships with the convergence horizon. These results collectively show that the mapping from initialization-time statistics to $T_{\text{conv}}$ is strongly non-linear, making ensemble regressors a more suitable choice. Accordingly, the main CAPE model uses a Random Forest regressor.

Table 3: Ablation of regressor choices for CAPE, CAPE (probe-only), and the scaling-law baseline. Lower MAE/RMSE and higher PearsonR indicate better performance.

| Method | Evaluation protocol | MAE | RMSE | PearsonR |
|---|---|---|---|---|
| **CAPE (full feature set)** | | | | |
| CAPE–Random Forest | Cross-fold (5-fold) | **4.63** | **8.10** | **0.89** |
| CAPE–Random Forest | Leave-one-dataset-out | **6.85** | **10.57** | **0.81** |
| CAPE–Random Forest | Leave-one-model-out | **7.27** | **11.04** | **0.79** |
| CAPE–Gradient Boosting | Cross-fold (5-fold) | 4.96 | 8.10 | 0.89 |
| CAPE–Gradient Boosting | Leave-one-dataset-out | 7.08 | 10.73 | 0.80 |
| CAPE–Gradient Boosting | Leave-one-model-out | 9.04 | 12.86 | 0.71 |
| CAPE–Ridge | Cross-fold (5-fold) | 8.39 | 11.84 | 0.75 |
| CAPE–Ridge | Leave-one-dataset-out | 9.99 | 13.86 | 0.66 |
| CAPE–Ridge | Leave-one-model-out | 15.22 | 19.58 | 0.38 |
| **CAPE (probe-only: $\log G^2, \log \tau$)** | | | | |
| Probe-only–Random Forest | Cross-fold (5-fold) | **13.10** | **21.33** | **0.05** |
| Probe-only–Random Forest | Leave-one-dataset-out | **13.90** | **21.69** | **-0.09** |
| Probe-only–Random Forest | Leave-one-model-out | **15.12** | **23.12** | **-0.16** |
| Probe-only–Gradient Boosting | Cross-fold (5-fold) | 13.31 | 21.03 | $-0.01$ |
| Probe-only–Gradient Boosting | Leave-one-dataset-out | 13.44 | 20.92 | $-0.13$ |
| Probe-only–Gradient Boosting | Leave-one-model-out | 13.94 | 21.71 | $-0.18$ |
| Probe-only–Ridge | Cross-fold (5-fold) | 13.30 | 21.01 | $-0.08$ |
| Probe-only–Ridge | Leave-one-dataset-out | 13.54 | 21.26 | $-0.27$ |
| Probe-only–Ridge | Leave-one-model-out | 13.16 | 18.66 | $-0.31$ |
| **Scaling-law baseline ($\log P, \log N$)** | | | | |
| Scaling-law–Ridge | Cross-fold (5-fold) | **11.87** | **17.23** | **0.28** |
| Scaling-law–Ridge | Leave-one-dataset-out | **11.81** | **17.29** | **0.27** |
| Scaling-law–Ridge | Leave-one-model-out | **12.34** | **17.82** | **0.18** |
| Scaling-law–Random Forest | Cross-fold (5-fold) | 10.56 | 16.09 | 0.45 |
| Scaling-law–Random Forest | Leave-one-dataset-out | 11.74 | 16.98 | 0.35 |
| Scaling-law–Random Forest | Leave-one-model-out | 14.22 | 20.01 | 0.07 |
| Scaling-law–Gradient Boosting | Cross-fold (5-fold) | 10.55 | 16.10 | 0.45 |
| Scaling-law–Gradient Boosting | Leave-one-dataset-out | 11.73 | 16.96 | 0.35 |
| Scaling-law–Gradient Boosting | Leave-one-model-out | 14.18 | 19.96 | 0.07 |

**Regressor choice for the scaling-law baseline.** The scaling-law baseline uses only two static features, $\{\log P, \log N\}$, which induce a nearly linear relationship with convergence steps. While tree-based regressors sometimes achieve slightly better performance in CV/LODO, the *LOMO* regime whose goal is to assess cross-model generalization—shows that Ridge achieves: (i) the lowest RMSE, (ii) the highest Pearson correlation, and (iii) the most stable behavior across architectures. Because the scaling-law baseline is meant to represent a *structurally simple, parametric predictor*, ridge regression is the most faithful and robust choice for this baseline, consistent with prior scaling-law literature and supported empirically by its superior LOMO performance.

