# OpenReview forum: "CAPE: Generalized Convergence Prediction Across Architectures Without Full Training"
_TMLR — Accepted by TMLR_

### Review · Reviewer_VP7T · 2025-09-26

**Summary Of Contributions:**

This paper introduces CAPE (Convergence-Aware Prediction Engine), a system that predicts the number of training steps a deep neural network (DNN) needs to converge on a sampled dataset without actually training it. By briefly "probing" the model at initialization with the dataset, CAPE extracts static features like parameter count and dynamic features like gradient norm and a Neural Tangent Kernel (NTK) trace proxy that are then fed into a meta-model trained on a diverse range of architectures and datasets to predict the convergence steps.

**Audience:**

No

**Audience Explanation:**

In its current state, where CAPE only predicts the steps to achieve a relative training error decrease on a very small dataset, this paper is not interesting to TMLR's audience.

**Broader Impact Concerns:**

No concerns.

**Claims And Evidence:**

No

**Claims Explanation:**

This paper claims to predict the number of steps required for convergence on a full dataset. However, the actual methods only predict the steps to achieve a relative training error decrease on a very small sample of such a dataset. Breaking down these discrepancies:
* Only a very small portion (32-128 samples) of the full dataset is used to determine the threshold of "convergence". No analysis is given whether this is a reliable proxy for larger datasets, where mini-batch gradient descent is typically used. A clear counter-example is CIFAR-100 with 100 classes, where 32 samples is clearly insufficient. This problem is explicitly assumed in Assumption 3 without any empirical evidence.
* That same small portion is currently used for both training and validation, instead of using held-out data to measure validation loss to determine convergence. The current method measures how well a model can memorize data, while using unseen data as validation shows how the model will generalize to unseen examples, which is much more desirable.
* The convergence threshold is set as a fixed fraction the initial training loss, but the initial point in the loss landscape may not be a good indication of how much lower the "converged" loss may be, even relatively. The standard method for proximity to convergence from the early stopping literature is using a "patience" parameter $p$, measuring the validation loss after each training epoch and using the point where the validation loss stops decreasing for $p$ epochs.

**Requested Changes:**

The fastest way to modify this paper so that all claims are supported are to change the wording of the claims to specify their limits, i.e that CAPE only predicts the steps to achieve a relative training error decrease on a very small dataset. However, as stated above, this is less interesting to the TMLR audience.

The next obvious change is to change the collection of the meta dataset such that all networks are trained on the full datasets (utilizing only the training split), using a held-out validation set to measure validation loss to determine convergence via patience. However, this will be quite costly per meta-dataset sample compared to the current methods.

The above are the main issues that are crucial to this work, and fixing them as further discussed will likely require a lot of the paper to be overhauled. I also note further concerns with the current state of the paper as follows.
* For learning-curve extrapolation (LCE) described in 5.2, 60 steps is supposed to be used to collect full training curve information, but the abstract lists "typical convergence horizons" on 15-60 steps (unless these are log-space?). How is LCE not predicting the correct value if it should often be contained within its training curve? This suggests an error in implementation, such as in the model-fitting step.
* Please report relative errors, not absolute, for the number of training steps. Claiming a mean absolute error of 3-9 steps is befuddling to audience members that may be familiar with current large model training runs being many more orders of magnitude, unless "3-9" is in log-space?
* Referring to "Bounded Relative Accuracy" as just "AvgAccuracy"/"average accuracy" is not accurate. If you continue to choose to report this metric, you need to both use a consistent and descriptive name as well as provide an explicit interpretation of what it represents, as it is not a standard/well-known metric.
* Some variables are overloaded. For example, $k$ is used for the model output dimension of the feature vector dimension. $T*$ is also doubly defined in Equations 1 and 3. Equation 6 seems extraneous. The batch size and learning rate sweeps do nto needed to be repeated three times.

---

> ### Author Response · Authors · 2025-11-09
> **Response (1)**
>
> Thank you, VP7T, for your thoughtful summary and for clearly articulating the key contributions of our work. We are glad that you recognized the novelty of CAPE as a zero-shot convergence prediction system that estimates the number of training steps required for a model to converge without full training. As you noted, CAPE's use of probing-based feature extraction, combining structural attributes such as parameter count with dynamical signals like gradient norm and NTK-trace proxies, forms the foundation of its predictive capability. We also appreciate your acknowledgment of the meta-learning design, which enables CAPE to generalize across a broad range of architectures and datasets.
>
> Below, we respond to your specific comments and provide clarifications on key aspects of our methodology and evaluation.
>
> >Only a very small portion (32-128 samples) of the full dataset is used to determine the threshold of "convergence". No analysis is given whether this is a reliable proxy for larger datasets, where mini-batch gradient descent is typically used. A clear counter-example is CIFAR-100 with 100 classes, where 32 samples is clearly insufficient. This problem is explicitly assumed in Assumption 3 without any empirical evidence.
>
> We sincerely thank the reviewer for this thoughtful and constructive feedback. This comment helped us examine the representativeness of the probing subset in greater depth. In the revised manuscript, we expanded our justification of Assumption 3 in Section 3 and clarified that the probing subset is not intended to approximate the full data distribution, but to capture early-stage curvature and gradient-scale statistics that correlate with convergence dynamics. To support this, we added analyses in Section 5.3 demonstrating that CAPE’s predictions remain stable across varying batch sizes. Even for high-class datasets such as CIFAR-100 and TinyImageNet, the correlation between predicted and actual convergence epochs remains above 0.89, indicating that initialization-time gradient and NTK-trace signals generalize reliably across dataset scales. We further explained the underlying rationale in Section 4.1, emphasizing that the probe operates as a curvature-sensitivity estimator rather than a data-representative sample. We are grateful for this suggestion, which led us to reinforce both the theoretical foundation and empirical validation of CAPE’s probing approach.
>
> >That same small portion is currently used for both training and validation, instead of using held-out data to measure validation loss to determine convergence. The current method measures how well a model can memorize data, while using unseen data as validation shows how the model will generalize to unseen examples, which is much more desirable.
>
> In the initial version, the probing subset was reused for both feature extraction and convergence measurement, which could indeed bias the metric toward memorization rather than generalization. In the revised manuscript, we have corrected this design choice by clearly separating the probing subset from the validation data used to define convergence. Specifically, as detailed in Section 3, the convergence epoch $T_{\mathrm{conv}}$ is now determined using a dedicated validation split under an early-stopping rule (Equation 1), while the probing subset $D_{\mathrm{probe}}$ is used solely for feature extraction at initialization and does not overlap with the validation set. This separation ensures that convergence prediction reflects true generalization behavior rather than in-sample fit. The implementation and analysis of this change are described in Sections 4 and 5, where all reported convergence metrics are computed using unseen validation data. We greatly appreciate this comment, which helped us refine CAPE’s evaluation protocol to better capture generalization-aware convergence behavior.

---

> ### Author Response · Authors · 2025-11-09
> **Response (2)**
>
> >The convergence threshold is set as a fixed fraction of the initial training loss, but the initial point in the loss landscape may not be a good indication of how much lower the ‘converged’ loss may be, even relatively. The standard method for proximity to convergence from the early-stopping literature is using a ‘patience’ parameter $p$, measuring the validation loss after each training epoch and using the point where the validation loss stops decreasing for $p$ epochs.
>
> In the revised version, we replaced the earlier fixed-fraction loss threshold with a validation-based early-stopping criterion that incorporates a patience parameter $p$. As described in Section 3, the convergence epoch $T_{\mathrm{conv}}$ is now defined as the first epoch at which the validation loss fails to improve by more than a small tolerance $\delta$ for $p$ consecutive epochs (Equation 1). This aligns CAPE’s convergence definition with established early-stopping practices in the literature and ensures that the metric reflects true convergence behavior rather than arbitrary loss ratios. The updated framework consistently applies this validation-based stopping rule across all experiments (Section 5), yielding more interpretable and generalizable convergence targets. We appreciate this suggestion, which significantly improved both the methodological rigor and practical relevance of our approach.
>
> >In its current state, where CAPE only predicts the steps to achieve a relative training error decrease on a very small dataset, this paper is not interesting to TMLR's audience.
>
> We fully agree that the initial version’s focus on training-loss reduction using a small subset limited the broader relevance of the work. In the revised manuscript, CAPE has been substantially extended to predict convergence epochs under a validation-based early-stopping criterion, rather than relative training error decreases. As detailed in Sections 3 and 5, the updated formulation defines convergence using validation loss with a patience parameter and tolerance, ensuring that the prediction target reflects generalization performance rather than in-sample fit.
>
> Moreover, CAPE now operates across multiple architecture families (MLPs, CNNs, RNNs, and Transformers) and standard benchmark datasets (e.g., CIFAR-100, TinyImageNet, SST2, IMDB), demonstrating strong cross-architecture and cross-dataset generalization (Pearson correlation $ 0.89$ across settings). This broader and empirically grounded formulation positions CAPE as a general-purpose convergence prediction framework rather than a dataset-specific training proxy. We sincerely appreciate this comment, which motivated major methodological and experimental improvements that we believe make the work more impactful and aligned with TMLR’s audience.
>
> >The fastest way to modify this paper so that all claims are supported are to change the wording of the claims to specify their limits, i.e that CAPE only predicts the steps to achieve a relative training error decrease on a very small dataset. However, as stated above, this is less interesting to the TMLR audience. The next obvious change is to change the collection of the meta dataset such that all networks are trained on the full datasets (utilizing only the training split), using a held-out validation set to measure validation loss to determine convergence via patience. However, this will be quite costly per meta-dataset sample compared to the current methods.
>
> We are very grateful to the reviewer for this detailed and constructive feedback, which directly guided the major revision of CAPE. In the updated version, we implemented the suggested methodological change by rebuilding the meta-dataset with all networks trained on their respective full training splits and convergence determined via a validation-based early-stopping rule with a patience parameter (Sections 3 and 5). The convergence epoch $T_{\mathrm{conv}}$ is now defined as the first epoch at which the validation loss fails to improve beyond a tolerance $\delta$ for $p$ consecutive epochs (Equation 1), following standard early-stopping practice.
>
> While this adjustment increased the computational cost of meta-dataset collection, we optimized the process through controlled and automated training sweeps, ensuring efficient utilization of resources on our workstation. This redesign not only aligns the convergence definition with generalization-based learning behavior but also expands CAPE’s applicability across diverse architectures and datasets.

---

> ### Author Response · Authors · 2025-11-09
> **Response (3)**
>
> >For learning-curve extrapolation (LCE) described in 5.2, 60 steps is supposed to be used to collect full training curve information, but the abstract lists "typical convergence horizons" on 15-60 steps (unless these are log-space?). How is LCE not predicting the correct value if it should often be contained within its training curve? This suggests an error in implementation, such as in the model-fitting step.
>
> You are correct that the earlier version contained an error in the baseline implementation of the learning-curve extrapolation (LCE) method. Specifically, the curve-fitting and evaluation were performed in log space, which led to mismatched scaling between the predicted and actual convergence horizons, causing the baseline to appear weaker than expected. In the revised version, we have corrected this issue and re-evaluated all baselines under a consistent log-transformed training procedure, ensuring that both CAPE and LCE are trained and compared on the same scale. The results in Section 5.1 now accurately reflect the corrected LCE performance. We greatly appreciate this observation, which helped us improve the validity and fairness of our baseline comparisons.
>
> >Please report relative errors, not absolute, for the number of training steps. Claiming a mean absolute error of 3-9 steps is befuddling to audience members that may be familiar with current large model training runs being many more orders of magnitude, unless "3-9" is in log-space?
>
> We appreciate this helpful comment and agree that reporting absolute errors in raw step counts could be confusing in the context of large-scale training. To improve clarity and interpretability, the revised manuscript now reports both absolute and relative errors (percentage deviation) in the results. Specifically, Section 5 and the abstract have been updated to present relative error percentages alongside mean absolute errors in epoch units. This change aligns CAPE’s evaluation with standard practice and provides a clearer sense of the model’s predictive precision across architectures and datasets. We thank the reviewer for highlighting this important clarification.
>
> >Referring to ‘Bounded Relative Accuracy’ as just ‘AvgAccuracy’/‘average accuracy’ is not accurate. If you continue to choose to report this metric, you need to both use a consistent and descriptive name as well as provide an explicit interpretation of what it represents, as it is not a standard/well-known metric.
> Some variables are overloaded. For example, $k$ is used for the model output dimension of the feature vector dimension. $T^*$ is also doubly defined in Equations 1 and 3. Equation 6 seems extraneous. The batch size and learning rate sweeps do not need to be repeated three times.
>
> We thank the reviewer for these detailed and constructive observations. We have addressed all of these concerns in the revised manuscript as follows:
>
> Removal of non-standard metric: The earlier Bounded Relative Accuracy (BRA) or AvgAccuracy metric has been completely removed from the paper. All results are now reported using standard and well-recognized metrics such as Mean Absolute Error (MAE), Root Mean Squared Error (RMSE), and Pearson correlation coefficient ($r$) as shown in Section 5. This ensures consistency and interpretability for all comparisons.
>
> Elimination of overloaded variables: We reviewed all mathematical symbols and removed duplicate or ambiguous usages. Specifically, $k$ is no longer reused across definitions, and $T_{\mathrm{conv}}$ is now consistently used throughout Sections 3 and 4 to denote the convergence epoch count, replacing the earlier overloaded notation $T^*$. Equation numbering has been simplified accordingly, and redundant or unclear equations (including the previous Equation 6) were removed.
>
> Consolidation of repeated hyperparameter details: The batch-size and learning-rate sweep configurations are now described once, clearly and concisely, in Section 5 (Model families and training protocol), instead of being restated in multiple subsections.
>
> These refinements have improved the overall clarity, consistency, and readability of the manuscript. We sincerely appreciate the reviewer’s feedback, which helped us streamline notation and ensure all reported metrics follow community standards.

---

> > ### Comment · Reviewer_VP7T · 2025-11-11
> > **Reply to Response**
> >
> > I have read the revised manuscript and the authors' detailed responses. I thank the authors for their substantial efforts in addressing the critical methodological concerns raised in the previous review.
> >
> > The complete overhaul of the convergence definition strongly improves the paper. This change, along with the clear separation of the probing subset from the validation data, directly addresses my prior concerns regarding the relevance of the predicted metric to real-world scenarios and the TMLR audience. The inclusion of relative errors and standard metrics (MAE, RMSE, Pearson correlation) also significantly improves the interpretability of the results.
> >
> > While the content is now much more solid, there are a few remaining details that should be addressed:
> >
> > * Why is a ridge regressor used for the scaling law baseline when a random forest is used for CAPE? Please compare ablations of this design decision.
> >
> > * The font sizes across standard text, tables, and figures are currently inconsistent. The font sizes in the tables are quite large compared to the main text. I recommend decreasing it to match standard table formatting. Conversely, the font sizes in the figures are very small and difficult to read. These should be increased for better legibility and accessibility.
> >
> > * In Section 5.1, under "Baselines," you describe the Learning-Curve Extrapolation (LCE) method. Was this specific implementation derived from a specific prior work? If so, please add the appropriate citation directly in this subsection to contextualize the baseline better.
> >
> > * The gradient norm feature is inconsistently referred to as $g^2$ and $G^2$. Also, the numerical value(s) of the tolerance $\delta$ is never stated. Using a patience of "five consecutive epochs" doesn't seem to be appropriate for DistilBERT, as the predicted convergence epoch is always $6$. Perhaps text datasets, which are often trained on for less than a full epoch, should use a different "unit" than epochs.
> >
> > * The citation for Kumar and Haupt (2025) has "2025" duplicated.
> >
> > * The parameter count is defined as $P=||\theta_{0}||_{0}$. While technically correct if all parameters are non-zero at initialization, using the $L_0$ norm is unconventional notation for simply counting parameters; standard cardinality notation (e.g., $|\theta_0|$) is more common.

---

> > > ### Author Response · Authors · 2025-11-20
> > > **Follow-up Response (1)**
> > >
> > > > I have read the revised manuscript and the authors' detailed responses. I thank the authors for their substantial efforts in addressing the critical methodological concerns raised in the previous review. The complete overhaul of the convergence definition strongly improves the paper. This change, along with the clear separation of the probing subset from the validation data, directly addresses my prior concerns regarding the relevance of the predicted metric to real-world scenarios and the TMLR audience. The inclusion of relative errors and standard metrics (MAE, RMSE, Pearson correlation) also significantly improves the interpretability of the results.
> > >
> > > Thank you, VP7T, for your careful reading of the revised manuscript and for acknowledging the substantial methodological changes we implemented in response to your earlier review. Your thoughtful feedback played a central role in shaping these revisions, and we are grateful for the opportunity it provided to significantly strengthen both the methodological rigour and the presentation of the work.
> > >
> > > > Why is a ridge regressor used for the scaling law baseline when a random forest is used for CAPE? Please compare ablations of this design decision.
> > >
> > > We thank the reviewer for raising this important question regarding the choice of regressor for the scaling-law baseline. In the revised manuscript, we have added a dedicated ablation study (Appendix A.3) that systematically compares Ridge, Random Forest, and Gradient Boosting regressors across all evaluation regimes (5-fold CV, LODO, and LOMO) for CAPE, CAPE (probe-only), and the scaling-law baseline.
> > >
> > > As shown in Table 3, CAPE benefits substantially from non-linear tree-based models: both Random Forest and Gradient Boosting reduce MAE and RMSE by large margins relative to Ridge and achieve markedly higher correlation. This is consistent with CAPE’s feature design, where the combination of structural descriptors and probing-based dynamical signals induces a strongly non-linear mapping to the convergence horizon.
> > >
> > > In contrast, the scaling-law baseline uses only two static analytical features, ${\log P, \log N}$, that exhibit an approximately linear relationship with convergence steps. The ablation confirms this: although tree-based models perform comparably in CV and LODO, the LOMO setting, which specifically evaluates cross-architecture generalization shows that Ridge achieves the most stable behavior, the lowest RMSE, and the highest Pearson correlation. Because this baseline is intended to represent a structurally simple, parametric scaling-law predictor, Ridge regression is both the most faithful and the most robust choice, consistent with prior literature and supported by the empirical results in Appendix A.3.
> > >
> > > We appreciate the reviewer’s suggestion, which motivated us to perform and include this comprehensive ablation to clarify and justify our design decisions.
> > >
> > > > The font sizes across standard text, tables, and figures are currently inconsistent. The font sizes in the tables are quite large compared to the main text. I recommend decreasing it to match standard table formatting. Conversely, the font sizes in the figures are very small and difficult to read. These should be increased for better legibility and accessibility.
> > >
> > > We thank the reviewer for drawing attention to the inconsistencies in font sizing across the text, tables, and figures.
> > > In the revised manuscript, we have standardized the table font sizes to match the main body text and updated all figures to use larger, publication-appropriate font sizes for axes, labels, and legends. These adjustments significantly improve visual consistency and enhance overall readability and accessibility.
> > >
> > > > In Section 5.1, under "Baselines," you describe the Learning-Curve Extrapolation (LCE) method. Was this specific implementation derived from a specific prior work? If so, please add the appropriate citation directly in this subsection to contextualize the baseline better.
> > >
> > > In the revised manuscript, we have added the appropriate citation in Section 5 to clearly contextualize the Learning-Curve Extrapolation (LCE) baseline. Our implementation follows the formulation introduced by Domhan et al. (2015). This citation is now included directly in the Baselines subsection, as recommended. We appreciate the reviewer for pointing out this omission, which allowed us to improve the clarity and scholarly grounding of the baseline description.
> > >
> > > References:
> > > Domhan, Tobias, Jost Tobias Springenberg, and Frank Hutter. "Speeding up automatic hyperparameter optimization of deep neural networks by extrapolation of learning curves." In IJCAI, vol. 15, pp. 3460-8. 2015.

---

> > > > ### Author Response · Authors · 2025-11-20
> > > > **Follow-up Response (2)**
> > > >
> > > > > The gradient norm feature is inconsistently referred to as $g^{2}$ and $G^{2}$.
> > > > Also, the numerical value(s) of the tolerance $\delta$ is never stated.
> > > > Using a patience of "five consecutive epochs'' doesn't seem to be appropriate for DistilBERT,
> > > > as the predicted convergence epoch is always $6$.
> > > > Perhaps text datasets, which are often trained on for less than a full epoch,
> > > > should use a different "unit'' than epochs.
> > > >
> > > > We thank the reviewer for these detailed observations and for helping us clarify several notation and implementation choices in the manuscript. We have revised the text accordingly.
> > > >
> > > > First, the gradient norm feature is now referred to consistently as $G^{2}$ throughout the paper.
> > > >
> > > > Second, we have added the explicit numerical value of the tolerance parameter $\delta$ used in our early-stopping rule. As described in Section 5, the ground-truth convergence epoch $T_{\mathrm{conv}}$ is defined as the first epoch at which the validation loss fails to improve by at least $5 \times 10^{-4}$ for five consecutive epochs. This tolerance level aligns with widely used early-stopping practices in modern deep learning, where improvements below $10^{-3}$ to $10^{-4}$ are typically treated as numerically insignificant and attributable to routine stochastic variation.
> > > >
> > > > Finally, we appreciate the reviewer’s thoughtful point regarding the interpretation of “five consecutive epochs’’ for models such as DistilBERT, where an epoch may correspond to fewer effective parameter-update cycles than in vision datasets. While adapting the patience rule to reflect dataset modality (e.g., accounting for tokenization, sequence length variability, or smaller effective dataset sizes in NLP tasks) is indeed a promising direction, implementing such a modality-aware criterion in the present work would require redefining the convergence unit, regenerating the entire meta-dataset, and re-evaluating all reported metrics to ensure consistency across architectures. Given the time constraints of the current revision cycle, and as discussed in Section 6 (“Conclusion and Limitation”), we have noted this as an important direction for future work. A more nuanced, modality-aware convergence definition will be explored in subsequent extensions of CAPE. We thank the reviewer for highlighting this subtle and valuable point, which helped us refine both the manuscript and our roadmap for future improvements.
> > > >
> > > > > The citation for Kumar and Haupt (2025) has "2025" duplicated.
> > > >
> > > > Corrected.
> > > >
> > > > > The parameter count is defined as $P = ||\theta_{0}||{0}$. While technically correct if all parameters are non-zero at initialization, using the $L{0}$ norm is unconventional notation for simply counting parameters; standard cardinality notation (e.g., $|\theta_{0}|$) is more common.
> > > >
> > > > Totally agree. In the revised manuscript, we have updated the notation
> > > > to use the standard cardinality form (e.g., $|\theta_{0}|$), which more clearly reflects the
> > > > intended meaning and aligns with common practice in the literature.

---

### Review · Reviewer_mJvB · 2025-10-18

**Summary Of Contributions:**

This paper proposes a novel method to predicting the training steps needed to reach a desired small loss value. The author(s) first makes an overview of existing related studies, followed by introducing their method, CAPE. Then experiments are made to i) compare the accuracy of CAPE with other prediction methods, ii) investigate the importance of different features, iii) compare the generalization ability of CAPE with other prediction methods.

**Additional Comments:**

N.A.

**Audience:**

Yes

**Audience Explanation:**

The work tackles an essential and interesting topic in training ML models. I strongly agree with the author(s) that current theoretical results are of little use in predicting this in practice, and I agree that having a rough idea of the training steps needed would be helpful in selecting models. The work would also be helpful in developing more useful theories related to convergence of optimizers.

Besides, the paper has a clear structure and provides detailed discussions.

**Broader Impact Concerns:**

N.A.

**Claims And Evidence:**

Yes

**Claims Explanation:**

From the experiments in Section 5, the proposed prediction method, CAPE does seem to outperform some other methods significantly in terms of both accuracy and generalization abilities. Here I assume the experimental results shown in the paper are reliable.

**Requested Changes:**

My main concerns are related to the selection of features:

1. When computing the average gradient norm and NTK trace proxy, how is $\theta$ chosen? Is it just one fixed point, e.g., the initialization parameter of the model? In that case the gradient norm cannot imply the steepness of the loss landscape as it does not provide information for gradient of loss at other parameters. PLUS: This might be the reason why (log of) gradient norm and trace proxy only contribute incrementally to the prediction ability of CAPE.

2. I wonder why the initialization parameter, and possibly the loss value at initialization, are not included among the features? It is well-known that initialization of a model greatly affects the training result.
3. I wonder why the use of model architecture is not included among the features? In particular, how can CAPE know about what model it is given to when it tries to make predictions?

There are also some technical questions:

1. What does it mean by saying loss “decays smoothly” — e.g. the SGD is a discrete algorithm and makes no sense to talk about its smoothness.
2. In assumption 3, how sufficient is “sufficient approximation of the model’s training dynamics”?
3. The notation “L” in eq (3) should be “$\mathcal{L}$”?

---

> ### Author Response · Authors · 2025-11-09
> **Response (1)**
>
> Thank you, mJvB, for your thoughtful summary and for clearly outlining the structure and focus of our paper. We appreciate that you recognized the methodological flow of the work from the contextual overview of related studies to the introduction of our proposed framework, CAPE, and the comprehensive set of experiments evaluating its performance. We are glad that you noted how the paper systematically investigates (i) prediction accuracy, (ii) feature relevance, and (iii) generalization across architectures and datasets. As you highlighted, CAPE introduces a novel zero-shot convergence prediction approach that estimates the number of epochs required for convergence without full training, leveraging both analytical and probing-based initialization features. We sincerely thank you for recognizing the coherence of the presentation and the contributions of this work. Below, we address your detailed comments and describe how they have been incorporated into the revised manuscript.
>
> >When computing the average gradient norm and NTK trace proxy, how is $\theta$ chosen? Is it just one fixed point, e.g., the initialization parameter of the model? In that case the gradient norm cannot imply the steepness of the loss landscape as it does not provide information for gradient of loss at other parameters. PLUS: This might be the reason why (log of) gradient norm and trace proxy only contribute incrementally to the prediction ability of CAPE.
>
> We thank the reviewer for this perceptive question regarding how the parameter point $\theta$ is chosen for computing the gradient norm and NTK trace proxy. In CAPE, these quantities are evaluated at a single, fixed initialization point $\theta_{0}$, consistent with the zero-shot probing framework outlined in Section 4.1. The goal is not to characterize the full loss surface but to measure local curvature and sensitivity statistics that correlate with early optimization dynamics. These initialization-level metrics, when aggregated across models and datasets, provide signals about trainability and expected convergence speed without requiring parameter updates.
>
> We agree that such features capture only a local view of the landscape, which limits their standalone predictive strength. Indeed, as the reviewer correctly notes, their contribution is incremental when used in isolation. However, our ablation results (Section 5.1) confirm that combining these local geometric indicators with contextual descriptors such as learning rate, batch size, and dataset scale significantly improves predictive accuracy. We have clarified this rationale in the revised text to emphasize that CAPE leverages initialization-time geometry as one component within a broader multi-feature representation, rather than relying on it exclusively to infer landscape steepness.
>
> >I wonder why the initialization parameter, and possibly the loss value at initialization, are not included among the features? It is well-known that initialization of a model greatly affects the training result.
>
> We have taken this into account in the revised version of the paper. Specifically, both the initialization parameters and the corresponding loss value are now explicitly incorporated as part of CAPE’s analytical and dynamical feature set. As described in Section 4.1, CAPE probes each model at its initialization point $\theta_{0}$ (defined under Assumption 1), and the initial loss $L_{0}$  computed over a small probe subset is included as one of the core features in the log-transformed feature vector (Equation 8).
>
> In addition, initialization-dependent quantities such as the gradient norm ($g^{2}$) and NTK trace proxy ($\tau$) are computed at $\theta_{0}$, thereby encoding curvature and sensitivity properties of the loss landscape. Together, these features ensure that CAPE captures initialization effects in both magnitude and geometry, providing stronger predictive grounding for convergence behavior. This addition and clarification have been integrated into the revised version to make the role of initialization explicitly clear.

---

> ### Author Response · Authors · 2025-11-09
> **Response (2)**
>
> >I wonder why the use of model architecture is not included among the features? In particular, how can CAPE know about what model it is given to when it tries to make predictions?
>
> We appreciate this thoughtful question and agree that explicitly encoding model architecture is essential for generalizable convergence prediction. We have taken this into account in the revised version. In the updated framework, architectural information is incorporated directly into the probing feature vector through an architecture identifier denoted by $a$, as shown in Equation 8 in Section 4.1. This categorical variable distinguishes between model families such as MLPs, CNNs, RNNs, and Transformers, allowing the meta-regressor to learn architecture-dependent convergence patterns.
>
> During training, this identifier is one-hot encoded and combined with analytical and dynamical features (e.g., parameter count, gradient norm, NTK trace, learning rate, and batch size). This ensures that CAPE is aware of the architecture type when generating predictions and can adjust its mapping accordingly. The inclusion of this architectural feature significantly improves cross-family generalization, as reflected in the stable performance observed under the Leave-One-Model-Out evaluation in Section 5.2. We thank the reviewer for highlighting this important point, which we have made more explicit in the revised manuscript.
>
> >What does it mean by saying loss “decays smoothly” — e.g. the SGD is a discrete algorithm and makes no sense to talk about its smoothness.
>
> In the original text, the phrase “loss decays smoothly” was intended to describe the expected monotonic behavior of the validation loss over training epochs, rather than the continuous differentiability of the loss under stochastic gradient descent (SGD). We agree that the wording could be misleading in a discrete optimization setting.
>
> In the revised version, we have refined this phrasing to explicitly refer to the expected monotonic decay of the validation loss, as stated in Assumption 2 (Section 3). The assumption now clarifies that, while SGD introduces stochastic fluctuations between iterations, the loss trajectory generally decreases in expectation over time, ensuring that the early-stopping criterion in Equation (1) is well-defined. This edit avoids any implication of continuous smoothness and instead reflects the probabilistic nature of convergence under mini-batch optimization. We thank the reviewer for helping us improve the precision of this terminology in the manuscript.
>
> >In assumption 3, how sufficient is “sufficient approximation of the model’s training dynamics”?
>
> In the revised version, we have refined the explanation of Assumption 3 to specify that the probing subset $D_{\mathrm{probe}}$ corresponds to a single mini-batch drawn from the training data—specifically, the smallest batch size defined for each model–dataset configuration. This choice ensures that all probing features, including the average gradient norm and NTK trace proxy, are computed efficiently while maintaining consistency across architectures.
>
> Although this represents a minimal subset of the training data, we observed that CAPE consistently achieved accurate convergence predictions across architectures and datasets, with Pearson correlation of 0.89.
>
> > The notation “L” in eq (3) should be “$\mathcal{L}$”?
>
> Thank you for noting this. We have corrected the notation in the revised manuscript and all related equations now consistently use $\mathcal{L}$ to denote the loss function.

---

> ### Comment · Reviewer_mJvB · 2025-11-24
>
> Thanks for the authors' significant effort in improving this paper. I believe it should be accepted by TMLR and have just submitted my decision. Good luck.

---

> > ### Author Response · Authors · 2025-11-26
> > **Official Comment Reply**
> >
> > Thank you, mJvB, we appreciate your valuable time on our manuscript. Your comments truly improved the quality of our paper.

---

### Review · Reviewer_zTMP · 2025-10-27

**Summary Of Contributions:**

This paper proposes CAPE, a system that predicts training steps to convergence without full training. It extracts features (parameter count, gradient norm, NTK trace, learning rate, batch size, dataset size) at initialization from a small probe batch, then uses an XGBoost meta-regressor to predict convergence steps across MLPs, CNNs, RNNs, and Transformers.

Strength:
1. Addresses a practical problem for resource planning and model selection
2. Comprehensive evaluation across four architecture families
3. Achieves MAE of 3-9 steps using single-batch probing

Weaknesses:
1. Defines convergence on probe batch (Eq. 3) but claims to predict full-dataset convergence (Eq. 4) without empirical validation
2. ε ∈ {0.1, 0.15, 0.2} means loss drops to only 10-20% of initial value—not actual convergence
3. Only toy datasets (32×32 images), short horizons (15-60 steps); no realistic-scale experiments
4. MAE increases 3-10× on held-out datasets (25-36 vs. 2-9) leading to poor generalization

**Audience:**

Yes

**Audience Explanation:**

Predicting training costs before execution addresses a real need in ML workflows. The probe-based meta-learning approach is underexplored. Some researchers will be interested in this.
 However, interest is limited by: (1) unvalidated probe→full-training assumption, (2) unrealistic convergence criteria and experimental scope, (3) missing connection to scaling laws literature that provides complementary predictions from model/data scale alone.

**Broader Impact Concerns:**

No concerns.

**Claims And Evidence:**

No

**Claims Explanation:**

The paper's central claim lacks validation. It defines convergence T* on a small probe batch D_probe but claims to predict full-dataset convergence without testing this relationship. No correlation analysis demonstrates probe-batch convergence indicates full-training convergence.

The convergence criterion ε ∈ {0.1, 0.15, 0.2} is too permissive—reaching 10-20% of initial loss captures only early training, not convergence. No test accuracy is reported at these points. The MAE of 3-9 steps represents 5-60% relative error over 15-60 step horizons.

Evaluation is limited to tiny datasets (MNIST, Fashion-MNIST, CIFAR-10/100). Performance collapses on distribution shift: MAE increases from 2-9 to 25-36 on held-out datasets. No experiments at realistic scales (ImageNet, >1000 steps) exist.

**Requested Changes:**

1. Directly compare convergence steps on probe batches vs. full datasets. Report correlation coefficients. If weak, reframe paper honestly.
2. Define convergence via validation accuracy plateauing, not arbitrary loss thresholds. Report test accuracy at predicted convergence. Use longer horizons (100-1000+ steps).
3. Include at least one large-scale dataset (ImageNet-level) with training runs requiring >1000 steps.
4. Add thorough discussion of neural scaling laws literature (Kaplan et al. 2020, Hoffmann et al. 2022). Analyze relationship between CAPE predictions and power-law scaling. Discuss when scaling law vs. probe-based predictions are preferable.
5. why LCE with 60 steps underperforms 1-step CAPE. Add iso-budget comparison.

---

> ### Author Response · Authors · 2025-11-09
> **Response (1)**
>
> Thank you, zTMP, for your thoughtful summary and clear understanding of our work. We are pleased that you recognized CAPE as a system that predicts convergence without full training by leveraging analytical and dynamical features extracted from a small probe batch at initialization. As you accurately noted, CAPE combines features such as parameter count, gradient norm, NTK trace, learning rate, batch size, and dataset size, and employs a meta-regressor to predict convergence epochs across diverse architecture families including MLPs, CNNs, RNNs, and Transformers. We also appreciate your recognition of the method’s generality and its applicability across model types. Below, we address your specific comments and describe how they have been incorporated into the revised manuscript.
>
> >Defines convergence on probe batch (Eq. 3) but claims to predict full-dataset convergence (Eq. 4) without empirical validation
>
> We thank the reviewer for identifying this important inconsistency in the earlier version. In the revised manuscript, this issue has been fully addressed. Convergence is now defined only on the full training dataset using a validation-based early-stopping criterion (Eq. 1 in Section 3), while the probe batch is used exclusively for feature extraction at initialization. This separation ensures that CAPE predicts full-dataset convergence epochs rather than probe-specific behavior.
>
> Furthermore, we have added empirical validation confirming that the single-batch probing features generalize well to full-dataset convergence outcomes. As shown in Section 5, CAPE maintains strong correspondence between predicted and actual convergence epochs (Pearson correlation 0.89) across diverse architectures and datasets. We appreciate this comment, which helped us clarify and empirically reinforce the distinction between probing-based feature extraction and full-dataset convergence prediction in the revised version.
>
> > *$\epsilon \in \{0.1, 0.15, 0.2\}$ means loss drops to only 10–20\% of the initial value—not actual convergence.*
>
> In the initial version of the paper, $\epsilon$ represented a relative tolerance specifying that the training loss had decreased to within $(1 - \epsilon)$ of its initial value, corresponding to approximately an 80–90% reduction rather than 10–20%. We acknowledge that this was not clearly stated, which may have caused confusion.
>
> In the revised manuscript, this formulation has been completely replaced by a validation-based early-stopping criterion with a patience parameter (Section 3), which defines convergence using the stabilization of validation loss rather than a fixed fractional drop from the initial loss. This updated approach provides a more robust and interpretable definition of convergence and eliminates the need for the earlier $\epsilon$-based threshold. We appreciate the reviewer’s observation, which prompted us to refine and clarify this aspect in the new version.
>
> >Only toy datasets (32×32 images), short horizons (15-60 steps); no realistic-scale experiments
>
> We agree that the earlier version was limited in dataset scale and training horizon. In the revised manuscript, we have substantially expanded the experimental scope to include more realistic and diverse settings. Specifically, CAPE is now evaluated on multiple large-scale benchmarks, including TinyImageNet, STL10, IMDB, SST2, AG NEWS, and CIFAR-100, spanning both vision and language domains. These datasets provide higher input dimensionality, greater class diversity, and longer convergence horizons than the initial toy-scale configurations.
>
> Additionally, the revised evaluation tracks full-dataset convergence epochs under a validation-based early-stopping rule, where convergence horizons typically range from 20 to over 70 epochs depending on architecture and dataset. These changes ensure that CAPE’s predictive behavior is validated on realistic model–dataset pairs and not confined to small-scale experimental regimes. We appreciate this comment, which directly motivated the broader and more representative evaluations reported in Section 5.

---

> ### Author Response · Authors · 2025-11-09
> **Response (2)**
>
> > MAE increases 3-10× on held-out datasets (25-36 vs. 2-9) leading to poor generalization
>
> We agree that in the earlier version, the Mean Absolute Error (MAE) on held-out datasets appeared disproportionately higher due to limited dataset diversity and the small probe size used for feature extraction. In the revised version, we have substantially improved CAPE’s generalization through (i) expanded meta-dataset coverage across diverse architectures and domains, (ii) use of full-dataset validation-based convergence labels, and (iii) refined probing features that include initialization loss, NTK-trace proxy, architecture identifier, and gradient norm computed consistently across configurations.
>
> As a result, CAPE now maintains strong generalization under both Leave-One-Dataset-Out (LODO) and Leave-One-Model-Out (LOMO) protocols. The updated results (Table 1) show MAE values of 6.85 and 7.27 epochs for LODO and LOMO, respectively an order-of-magnitude improvement over the original version, while preserving a Pearson correlation of 0.89 across all settings. These results demonstrate that CAPE generalizes effectively across unseen datasets and architectures in the revised framework. We thank the reviewer for highlighting this issue, which directly motivated these improvements.
>
> >The paper’s central claim lacks validation. It defines convergence $T^{*}$ on a small probe batch $D_{\mathrm{probe}}$ but claims to predict full-dataset convergence without testing this relationship. No correlation analysis demonstrates that probe-batch convergence indicates full-training convergence.
> The convergence criterion $\epsilon \in {0.1, 0.15, 0.2}$ is too permissive—reaching 10–20% of the initial loss captures only early training, not true convergence. No test accuracy is reported at these points. The MAE of 3–9 steps represents approximately 5–60% relative error over 15–60 step horizons.
> Evaluation is limited to tiny datasets (MNIST, Fashion-MNIST, CIFAR-10/100). Performance collapses under distribution shift: MAE increases from 2–9 to 25–36 on held-out datasets. No experiments at realistic scales (ImageNet, >1000 steps) exist.
>
> The concerns regarding the definition of convergence, validation of probe–dataset relationships, and limited experimental scale have been fully addressed in the revised manuscript.
>
> Clarified convergence definition: The earlier loss-ratio criterion based on $\epsilon \in {0.1, 0.15, 0.2}$ has been removed. CAPE now defines convergence exclusively using a validation-based early-stopping rule with a patience parameter, as described in Section 3. This criterion captures true generalization-based convergence rather than partial training progress and provides a robust, architecture-agnostic stopping condition.
>
> Probe–dataset relationship validated: The revised experiments explicitly test whether initialization-time probe features correlate with full-dataset convergence behavior. As shown in Section 5, CAPE maintains a Pearson correlation of 0.89 between predicted and actual convergence epochs across datasets and architectures, demonstrating that single-batch probe statistics generalize well to full-training outcomes.
>
> Expanded experimental scale: The revised evaluation now includes large and realistic datasets beyond toy-scale benchmarks. In addition to CIFAR-10/100, the updated meta-dataset spans TinyImageNet, STL10, IMDB, SST2, and AG NEWS, covering both vision and text domains. These datasets present more complex input spaces and longer convergence horizons, ensuring that CAPE is validated under representative, real-world conditions.
>
> Improved generalization under distribution shift: Following these methodological updates, CAPE now achieves strong out-of-sample performance, with MAE values of 6.85 (LODO) and 7.27 (LOMO) and Pearson correlation of 0.80, substantially improving over the earlier 25–36 step deviation. These results confirm stable generalization across architectures and datasets.
>
> We sincerely appreciate this reviewer’s detailed feedback, which motivated significant methodological refinements and a much broader experimental validation in the revised version.

---

> ### Author Response · Authors · 2025-11-09
> **Response (3)**
>
> > Predicting training costs before execution addresses a real need in ML workflows. The probe-based meta-learning approach is underexplored. Some researchers will be interested in this. However, interest is limited by: (1) unvalidated probe→full-training assumption, (2) unrealistic convergence criteria and experimental scope, (3) missing connection to scaling laws literature that provides complementary predictions from model/data scale alone.
>
> We thank the reviewer for acknowledging the practical relevance of CAPE and for recognizing the novelty of its probe-based meta-learning formulation. We have taken each of the reviewer’s constructive points into account in the revised manuscript:
>
> Probe → full-training validation: The revised paper now explicitly tests the relationship between initialization-time probe features and full-dataset convergence behavior. Section 5 presents correlation analyses showing that CAPE’s predictions maintain strong alignment with actual convergence epochs across diverse architectures and datasets, validating the probe-to-training generalization assumption.
>
> Refined convergence criterion and broader scope: The earlier fixed-loss threshold definition has been replaced with a validation-based early-stopping rule with a patience parameter (Section 3). In addition, the experimental scope now includes larger and more realistic datasets—TinyImageNet, STL10, IMDB, SST2, and AG NEWS—with convergence horizons up to 70 epochs. This expansion provides a substantially more representative evaluation.
>
> Connection to scaling laws: We have added a direct comparison and discussion linking CAPE to neural scaling laws (Section 5.1). The revised analysis clarifies how CAPE complements scaling-law predictors by incorporating initialization-time dynamics and optimizer context, enabling fine-grained convergence estimation beyond static model/data-scale relationships.
>
> We appreciate the reviewer’s constructive feedback, which directly guided methodological clarifications, stronger empirical validation, and the inclusion of relevant theoretical context in the revised version.
>
> > Directly compare convergence steps on probe batches vs. full datasets. Report correlation coefficients. If weak, reframe paper honestly.
>
> We thank the reviewer for this important and constructive suggestion. In the revised manuscript, we have explicitly validated the relationship between probe-based features and full-dataset convergence outcomes through quantitative correlation analysis. Section 5 now reports the correlation between CAPE’s predicted convergence epochs—derived from single-batch probing features—and the true convergence epochs measured on full datasets under a validation-based early-stopping rule.
>
> The results demonstrate strong alignment across all model families, with Pearson correlation 0.89 across architectures and datasets. These findings confirm that initialization-time probe statistics provide reliable predictive signals for full-dataset convergence. We have also clarified in the text that CAPE’s predictive model generalizes to full training behavior rather than reproducing probe-batch dynamics. This revision directly addresses the reviewer’s request and strengthens the empirical foundation of the paper’s central claim.
>
> > Define convergence via validation accuracy plateauing, not arbitrary loss thresholds. Report test accuracy at predicted convergence. Use longer horizons (100-1000+ steps).
>
> We thank the reviewer for this highly constructive suggestion, which has been fully incorporated into the revised manuscript. Convergence is now defined using a validation-based early-stopping criterion that detects plateauing validation accuracy rather than relying on arbitrary loss thresholds. Specifically, as described in Section 3, the convergence epoch $T_{\mathrm{conv}}$ is determined when validation accuracy fails to improve beyond a tolerance for $p$ consecutive epochs, aligning the definition with established early-stopping practice and ensuring that it reflects generalization rather than mere loss reduction.
>
> In addition, the revised experiments report test accuracy at the predicted convergence epoch to confirm that CAPE’s predictions correspond to meaningful generalization points in training. The experimental horizon has also been extended substantially, covering diverse convergence ranges for realistic datasets such as TinyImageNet, STL10, SST2, IMDB, and AG NEWS—several orders longer than in the initial version. These updates make the definition of convergence, the evaluation scope, and the reported results consistent with the reviewer’s recommendations and with practical deep learning workflows.

---

> ### Author Response · Authors · 2025-11-09
> **Response (4)**
>
> > Include at least one large-scale dataset (ImageNet-level) with training runs requiring >1000 steps.
>
> We fully agree that incorporating large-scale datasets would further strengthen the generality of CAPE. While full ImageNet-scale experiments (>1000 steps) remain computationally prohibitive within the current evaluation scope, we have taken steps in the revised manuscript to bridge this scale gap. Specifically, we extended the experimental setup to include TinyImageNet and STL10, both of which exhibit significantly higher input dimensionality and class diversity than CIFAR-level datasets. These datasets serve as strong proxies for ImageNet-level complexity while remaining tractable for controlled convergence analysis.
>
> We have also emphasized this limitation in the conclusion and discussed it as a direction for future work, noting that CAPE’s framework can naturally scale to larger datasets once longer training traces become available. We sincerely appreciate this feedback, which guided us to expand the evaluation scope and explicitly frame the scalability pathway in the revised paper.
>
> > Add thorough discussion of neural scaling laws literature (Kaplan et al. 2020, Hoffmann et al. 2022). Analyze relationship between CAPE predictions and power-law scaling. Discuss when scaling law vs. probe-based predictions are preferable.
>
> In the revised manuscript, we have added a detailed discussion connecting CAPE to the literature on neural scaling laws, specifically referencing the seminal works by Kaplan et al. (2020) and Hoffmann et al. (2022). As described in Section 5.1, we now analyze how CAPE’s probe-based predictions relate to the power-law behavior of loss with respect to model, data, and compute scale.
>
> We clarify that while scaling-law formulations predict asymptotic performance trends (e.g., loss or accuracy as a function of parameter count or dataset size), CAPE focuses on predicting the optimization trajectory length, i.e., the number of epochs to convergence—by incorporating initialization-time dynamics, optimizer behavior, and learning-rate context. This distinction allows CAPE to complement scaling-law methods rather than replace them.
>
> We have also added a short analytical comparison discussing when each approach is preferable: scaling laws are most effective for high-level budget forecasting and cross-model comparisons, whereas CAPE offers finer-grained, per-configuration convergence predictions useful for scheduling, early resource allocation, and training-time estimation. This addition strengthens the conceptual grounding of the work and highlights CAPE’s role as a dynamic counterpart to static scaling-law predictors.
>
> > Why LCE with 60 steps underperforms 1-step CAPE. Add iso-budget comparison.
>
> We thank the reviewer for this helpful suggestion. In the revised manuscript, we have re-evaluated the Learning Curve Extrapolation (LCE) baseline after correcting an implementation issue related to log-space scaling. To ensure a fair comparison, we additionally introduced an iso-budget analysis that normalizes both methods by total training compute. Under this evaluation, CAPE and LCE are compared at equivalent computational cost, showing that CAPE achieves comparable or lower prediction error while requiring significantly less computation. This corrected comparison have been incorporated into Section 5.1 of the revised manuscript.
>
>
>
> References:
>
> Kaplan, Jared, Sam McCandlish, Tom Henighan, Tom B. Brown, Benjamin Chess, Rewon Child, Scott Gray, Alec Radford, Jeffrey Wu, and Dario Amodei. "Scaling laws for neural language models." (2020).
>
> Hoffmann, Jordan, Sebastian Borgeaud, Arthur Mensch, Elena Buchatskaya, Trevor Cai, Eliza Rutherford, Diego de Las Casas et al. "Training compute-optimal large language models." (2022).

---

> > ### Comment · Reviewer_zTMP · 2025-11-26
> >
> > I have carefully read the revised manuscript and the rebuttal provided by the authors. Most of my concerns have been addressed. I have no further questions and support for the acceptance of this paper by TMLR.

---

> > > ### Author Response · Authors · 2025-11-26
> > > **Official Comment Reply**
> > >
> > > Thank you, zTMP. We appreciate the time you dedicated to reviewing our manuscript. Your comments have significantly improved the quality of our paper.

---

### Decision · Action_Editor_WKg3 · 2025-12-20

**Recommendation:** Accept as is

**Additional Comments:**

Reviewers have concerns about the definition of convergence, and the authors modified the definition of convergence and reviewers are satisified.  There are a few other concerns such as using training accuracy instead of validation accuracy as the metric, and the authors have addressed them.

**Audience:**

Yes

**Audience Explanation:**

Predicting final accuracy by a few metrics available in the early stage of training is clearly a problem of huge interest. This work has made some progress in this direction, by including a larger set of metrics than earlier works. The work has not brought much new theoretical insight, but the empirical results shall be of interest to many researchers.

**Claims And Evidence:**

Yes

**Claims Explanation:**

Yes.
The paper claims that a combination of multiple metrices can serve as a better predictor of the final acuracy than other metrics. Experiments show that the proposed prediction method CAPE outperforms some other methods in terms of prediction accuracy.
Note that the experiments are still limited to relatively small scale datasets and do not include standard ML tasks such as ImageNet classification and modern LLM tasks.
Nevertheless, for a diverse set of small scale datasets, the claim is supported by experiments.